# Temporal clustering of disorder events during the COVID-19 pandemic

**Gian Maria Campedelli**[ID][1]*, **Maria R. D'Orsogna**[2,3]

**1** Department of Sociology and Social Research, University of Trento, Trento, Italy, **2** Department of Computational Medicine, University of California Los Angeles, Los Angeles, CA, United States of America, **3** Department of Mathematics, California State University, Northridge, Los Angeles, CA, United States of America

* gianmaria.campedelli@unitn.it

**Data Availability Statement:** The original dataset is freely and openly available to anyone at the ACLED dedicated website: https://acleddata.com/analysis/covid-19-disorder-tracker/. The processed datasets and code utilized in this paper are

## Abstract

The COVID-19 pandemic has unleashed multiple public health, socio-economic, and institutional crises. Measures taken to slow the spread of the virus have fostered significant strain between authorities and citizens, leading to waves of social unrest and anti-government demonstrations. We study the temporal nature of pandemic-related disorder events as tallied by the "COVID-19 Disorder Tracker" initiative by focusing on the three countries with the largest number of incidents, India, Israel, and Mexico. By fitting Poisson and Hawkes processes to the stream of data, we find that disorder events are inter-dependent and self-excite in all three countries. Geographic clustering confirms these features at the subnational level, indicating that nationwide disorders emerge as the convergence of meso-scale patterns of self-excitation. Considerable diversity is observed among countries when computing correlations of events between subnational clusters; these are discussed in the context of specific political, societal and geographic characteristics. Israel, the most territorially compact and where large scale protests were coordinated in response to government lockdowns, displays the largest reactivity and the shortest period of influence following an event, as well as the strongest nationwide synchrony. In Mexico, where complete lockdown orders were never mandated, reactivity and nationwide synchrony are lowest. Our work highlights the need for authorities to promote local information campaigns to ensure that livelihoods and virus containment policies are not perceived as mutually exclusive.

## 1 Introduction

On March 11, 2020, the World Health Organization declared the coronavirus outbreak a global pandemic. No country has been spared from the far-reaching impacts of the ensuing viral disease, termed COVID-19, which touched and transformed most, if not all, aspects of societal living. In the absence of a widely available vaccine, and in light of the many unknowns posed by the pandemic, governments responded with a wide array of measures aimed at avoiding large scale public health crises with varying degrees of resources, capacity, and resolve. These included restrictions on travel and mobility, social distancing and quarantine

available at: https://github.com/gcampede/covid19-protests.

**Funding:** MRD acknowledges support from the Army Research Office (W911NF-18-1-0345) (url: https://www.arl.army.mil/), and the National Science Foundation (DMS-1814090). (Url: https://www.nsf.gov/) The funders had no role in study design, data collection and analysis, decision to publish, or preparation of the manuscript.

**Competing interests:** The authors have declared that no competing interests exist.

requirements, imposing school and business closures, implementing large scale testing, prohibiting large crowd gatherings. Periods of more or less stringent orders alternated depending on the local ebbs and flows of the pandemic. In parallel, governments also increased spending to reduce the risk of permanent damage to their economies and to alleviate unemployment and poverty. Policymakers have been confronted with philosophical and moral questions centered on how to balance individual freedoms and preserving the economy while protecting the lives of vulnerable populations, and on whether utilitarian-outcome or kantian-deontological approaches should be taken [1]. In addition, especially at the onset of the pandemic, different jurisdictions adopted different strategies due to the limited scientific understanding of COVID-19, and without being able to fully anticipate the medium-term health, social and political implications of their decisions. Sweden for example chose not to enter a lockdown while the rest of Europe did; in the US emergency orders were implemented after non-uniform threshold levels were met across the country, resulting in significant variability from county to county [2]. Similarly, the recent announcement of the development of effective vaccines is marked by questions not only on how to optimally scale up production but also how to distribute vaccines in an ethical manner [3, 4].

While not perfect and difficult to evaluate, the above interventions for the most part helped prevent more widespread transmission and buffer worse economic fallouts [5–7]. By the same token, they have provoked concerns over becoming drivers of economic uncertainty and deprivation [8], reinforcing economic inequality [9, 10], engendering political distrust [11], and leading to long-term mental health consequences as a result of prolonged physical isolation and reduced human interaction [12]. Ultimately COVID-19 has caused what are poised to be long-lasting economic, financial, and social crises worldwide.

Traditional and social media have played a significant role in disseminating COVID-19 related information [13, 14]. They have helped turn best practices into habits, and offered personalized networking opportunities in times of solitude. However, politicization of the pandemic, the sharing of non-vetted yet sensational or controversial findings or hypothesis, and algorithmic manipulation by platform hosts, have allowed for conspiracy theories and misinformation to spread, causing anxiety and discontent [15–17].

All these ingredients—the emergence of a pandemic most were not prepared for, the difficulty for policymakers in make clear-sighted decisions that could be well received by a diverse set of stakeholders, the online spread of distorted information, personal fears and the need to connect to others—have arguably combined to spark pandemic-related social unrest throughout the globe as manifest by the numerous protests, riots and disorders recorded in many countries. Although these events have attracted media attention and ignited public debate, their characteristics and timing are poorly understood. Protests ignited by past outbreaks offer little insight into current societal unrest due to their comparatively limited spatio-temporal scope, and the lack of accurate data collection at the time they occurred.

In this paper we investigate the temporal distribution of civil disorders directly attributable to the 2020 coronavirus pandemic and, specifically, the temporal clustering and self-excitability of events. These "contagion" trends are recurrent in human activity and have been observed in criminal behavior [18, 19], terrorist attacks [20, 21], political shocks [22, 23], violent conflicts [24, 25] and collective protesting and rioting [26–28]. We hypothesize they also characterize the current COVID-19 protests worldwide.

Our analyses focus on the three countries that were most hit by protests, riots, and violent events directly related to the COVID-19 pandemic, namely India, Israel, and Mexico, as per data made available through the "COVID-19 Disorder Tracker" (CDT) initiative [29] curated by the "Armed Conflict Location & Event Data Project" (ACLED) [30]. While we recognize that peaceful protests are distinct from riots or episodes of violence against civilians, and that

they carry different features and theoretical implications [31], we do not discriminate among them in our analyses due to their shared, high-level etiological trigger: the COVID-19 pandemic. Note that concurrent with the unfolding of the COVID-19 crisis are other episodes of civil unrest whose primary source is not pandemic-related, most notably the Lives Matter protests that originated in Minneapolis in late May 2020 in the wake of George Floyd's murder and that persist at this time of writing. ACLED does not tally these events, and we do not include them in our work as these demonstrations arise primarily in response to racial injustice, although racial disparities in COVID-19 cases may have contributed to the unrest.

For each of the three countries we focus on, we study the temporal clustering and self-excitability of pandemic-related demonstrations on the national and subnational scales. We assume all events recorded by ACLED in a given country are part of the same underlying stream of events, whereas on the local level we construct mutually exclusive geographical clusters through $k$-means clustering. Temporal self-excitation is studied by applying the Hawkes process [32], a stochastic point process initially used to understand aftershocks in the vicinity of an earthquake epicenter, and later adapted to finance, cell signaling, and disease-spread. In relation to social phenomena, Hawkes process have also been applied to describe and analyze the distribution of violent events, such as terrorist attacks [33], gun violence [34], and gang-related crimes [35]. Hawkes processes are non-Markovian extensions of Poisson processes. The latter are memory-less with events following a random temporal distribution, whereas the likelihood of an event in a Hawkes process depends on past ones, leading to clustering, memory effects, and self-excitability. These features are well-suited to study human behaviors that appear to be patterned in time. We fit both Poisson and Hawkes processes to data within each cluster to verify if and how temporal clustering emerges at various subnational scales.

Our analysis reveals that temporal clustering of pandemic-related demonstrations is a common feature in all three countries. Despite variations in the temporal distribution of events and in the magnitude of self-excitability and reactivity across the three different contexts, Hawkes processes better capture the underlying dynamics present in the data compared to Poisson processes. Furthermore, we find that self-excitability at the national level appears to emerge as the convergence of subnational, cluster-based self-excitatory events, rather than as the result of a meso-level stream of events. Our results highlight the interplay between the national and subnational socio-political discourse in the emergence of large-scale disorders that may erupt in response to centralized decisions, with information easily channeled through social media and other communication networks, but may also manifest locally, with the global source of discontent becoming particularized to local grievances and sources of tension.

The remainder of this paper is organized as follows. In the Background section we briefly review the protesting and rioting literature in other contexts. We also provide a synthetic account of the evolution of the pandemic in each of the three countries under investigation. In the Materials and Methods section will describe the ACLED data we utilize and outline our methodology. The statistical outcomes of our inferential models, along with additional analyses on the temporal characteristics of events in each country and cluster, are presented in the Results section. Finally, in the Discussion and Conclusion section we discuss the relevance of our results and their broader significance.

## 2 Background

COVID-19-related social unrest has been observed in several countries in the form of protests, riots, and other demonstrations, often in response to virus-containment decisions imposed by governments. These decisions may have seemed too onerous, unfair, or the root cause of economic uncertainty. It is important to understand the characteristics of this unrest for two

reasons: one is preventative, as demonstrations always carry the risk of widespread contagion, given their often chaotic and crowded nature, and of devolving into uncontrolled, violent clashes with authorities and/or among protesters and counter-protesters. The other is more introspective, as these protests may be manifestations of latent societal discontent, predating the pandemic, invigorated by newly mandated restrictive policies.

Though unique in its global reach, the current pandemic does not represent a *unicum* in terms of the social unrest it has caused. History has repeatedly been marked by uprising and protests in response to social, political, and economic crises. In recent decades, social movements and collective action have mobilized citizens in demanding changes to political regimes, economic policies, and more respect for human rights. A vast academic literature has emerged as a result, to study why, how, when large scale peaceful and violent demonstrations occur and what their repercussions may be [36]. Sociological and social movement studies [37] have been central in describing various theoretical frameworks underlying social unrest. Political science studies have helped dissect the political conditions in which riots and protests flourish and anticipate possible micro– and macro–consequences such as conflicts, regime changes, and revolutions [38]. Criminological studies have concentrated on understanding how deviant behavior can emerge in non-peaceful assemblies and in outlining related policing strategies [39, 40]. Psychological studies have helped unveil the behavioral and cognitive mechanisms occurring in individuals when engaged in collective action [41, 42].

The last decades have also witnessed the increased use of quantitative methodologies to study disorder events of various nature that have been used across disciplines and that have helped uncover consistent patterns. At the end of the 1970s, T.J. Sullivan [26] and M.I. Midlarsky [27], while belonging to distinct academic fields (social psychology and political science, respectively), highlighted "universal" features in the size distribution and dynamical evolution of crowds. M.I. Midlarsky, in particular, emerged as a strong advocate for the incorporation of quantitative frameworks within the study of collective social phenomena. In one of his seminal works, he demonstrated the presence of two underlying processes driving the urban disorders of 1966-67 in several US cities, diffusion and contagion, at a time when neither had been proven. The same empirical techniques were later applied to the study of transnational terrorism [43].

The greater availability of data in recent years has vastly improved our understanding of societal unrest and has led to refined statistical and computational approaches. Sources include law enforcement and institutional organizations, social media platforms and news agencies [44–46]. Among the most investigated scenarios are the 2005 Paris riots [47, 48], the 2011 London riots [49, 50], political protests in Latin America [51] and the Arab Spring [52]. In all cases non-random behaviors were detected such as clustering processes, cascading dynamics, and self-excitability. Complementary theoretical propositions helped diagnose the causal mechanisms of the inferred patterns and include hierarchical patterns [53, 54], rational choice [50], identity theory [55].

In the spirit of expanding the extant literature on human dynamics during disorders, we focus on pandemic-related demonstrations in the three nations that exhibit the greatest number of events, India, Israel and Mexico. Although these countries vary greatly in terms of geographical, socio-political, and economic fabric, our data-driven study will show that self-excitability and clustering emerge across all three as fundamental features of pandemic-related protests and riots.

## 2.1 Overview of the pandemic in select countries

**2.1.1 India.** Currently, India is the country with the second-highest recorded number of COVID-19 cases worldwide, with more than 10 million infections, corresponding to roughly

12% of the global count [56]. The first case to be confirmed was a medical student returning from Wuhan, China to the state of Kerala who tested positive on January 30th 2020 [57]. India's first fatality was reported on March 12th, amidst growing numbers of infections. On March 22nd, Prime Minister Narendra Modi called for a 14-hour voluntary national lockdown, which became mandatory three days later, when only essential services were allowed to remain open. On April 14th the lockdown was extended to May 3rd and on April 29th India's death toll reached one thousand. On May 1st, the lockdown was extended for two more weeks although some flexibility was allowed, depending on the local spread of the virus. Other economic, financial and social measures were introduced as lockdown restrictions were extended through the end of May. Starting in June, containment policies were progressively eased. On August 30th, the country recorded the highest number of new cases worldwide, in excess of 78,000 [58]; this record was shattered on September 12th when India tallied more than 95,000 infections. Schools were partially reopened at the end of summer and on September 30th states and Union Territories were allowed local autonomy on certain pandemic-related matters.

Protests against the government's management of the COVID-19 crisis have been widespread and sustained. Migrant workers unable to return home due to lockdown orders staged protests at railway stations, health workers went on strike over the lack of protective equipment and adequate pay, agricultural workers demanded payment of lockdown wages and protested the lack of aid despite government promises, students demonstrated against holding University admission exams and requested they be deferred due to infection risks and the difficulty of reaching exam centers. General strikes were organized to rally against poverty and unemployment triggered by COVID-19. On some occasions, episodes of state repression and violent enforcement of lockdown orders occurred as an attempt to quell these disorders. Long standing conflicts tied to identity politics, immigration, secessionist movements and boundary disputes have also been impacted by COVID-19. Massive protests had engulfed the country in 2019 and early 2020 in response to the Citizenship Amendment Act (CAA) which would grant citizenship rights on the basis of religion [59]. Political and separatist demonstrations, including those related to the CAA, declined following lockdown orders but reignited in June as restrictions eased, adding to pandemic-related disorders. In India 50% of the total population had internet access in 2020 [60].

**2.1.2 Israel.** Israel recorded its first confirmed COVID-19 case on February 21st 2020, after the return of a female citizen quarantined on the Diamond Princess in Japan [61]. Authorities soon issued a 14-day isolation policy for citizens who had visited South Korea or Japan; social distancing and related measures were imposed on March 11th, a ban on public gatherings of more than 10 people followed on March 15th. Finally, on March 19th, a state of emergency was declared and a national lockdown imposed. A contact-tracing program was approved on March 16th, allowing the Israeli Security Agency to track the mobility of individuals through mobile phone data. The program quickly sparked controversy and nationwide protests over the invasion of privacy and the overreaching surveillance of citizens. It was halted at the end of April. Additional restrictions were imposed between the end of March, in anticipation of the Passover Seder festivity on April 8th including a travel ban and the creation of a restricted zone in Bnei Brak, an ultra-Orthodox town east of Tel Aviv with one of the highest rates of coronavirus cases in the country. Restrictions were eased between the end of April and the beginning of May. Retail stores were allowed to open on April 24th; school reopenings took place between May 3rd and May 19th. In July, in response to steadily increasing "second wave" infections, the Knesset reauthorized the contact-tracing program, once more igniting civil liberties organizations; new social distancing rules were also imposed. A "traffic light" monitoring plan was announced on August 31st whereby a color representative of risk levels would be assigned to all Israeli cities [62]. Each color would be associated to more or less restrictive

rules. On September 6[th] schools were closed and night-time curfews were imposed on forty high risk communities, including nine in Jerusalem, affecting approximately 1.3 million people. A new national lockdown was issued for September 18[th], concurrent with the Jewish High Holy Days, and further restrictions announced on September 23[rd]. On October 18[th] some restrictions were lifted, with further reopenings announced throughout November 2020, although some high impacted communities remained under lockdown. Finally, a "third wave" of infections was accompanied by a third nationwide lockdown imposed on December 27[th].

Protests in Israel have been common throughout the pandemic, with demonstrations against the government's handling of the crisis, and the alleged corruption of Prime Minister Benjamin Netanyahu. After restrictions were placed to curb demonstrations, nationwide protests were held on October 3[rd]. Clashes with police, rock-throwing and confrontations between protesters and anti-protesters were recorded [63]. In Israel 84% of the total population had internet access in 2020 [64].

**2.1.3 Mexico.**    On February 28[th] 2020, Mexico confirmed its first COVID-19 cases as three men who had traveled to Bergamo, Italy; the first COVID-19 related death was recorded on March 18[th]. On March 23[rd] President Andrès López Obrador unveiled a national campaign to promote social distancing, self-isolation, and other measures to contain the spread of the virus. On March 30[th], Mexico declared a state of health emergency, suspending all nonessential activities for a month. Plans for gradual reopening were announced on May 18[th], with the mayor of Mexico City presenting a blueprint for a "new normality" in the capital on May 20[th]. On June 1[st] the government imposed a traffic light monitoring plan at the state level, similar to the one described for Israel. Cases steadily increased starting from the end of May leading to overcrowded hospitals. On July 30[th] it was announced that governors who altered the traffic light status of their states would face criminal sanctions. Record new cases of more than 10,000 new infections were recorded on November 25[th].

A few weeks into the pandemic, several media outlets began reporting that drug cartels were distributing food and medical supplies to citizens; over time criticisms mounted on government's handling of the pandemic, the too modest spending on public heath and measures to help the economy, the many deaths in Mexico City, inadequate testing, medical personnel, waste disposal, and possible concealment of the real number of COVID-19 cases and fatalities. The president has been accused of having resisted full lockdown measures despite rising cases, to keep the economy open, drawing criticism from the WHO and the IMF. At this time of writing, Mexico is still among the most affected countries, with more than one fifth concentrated in its capital.

Labor groups have protested the government's inadequate support of the population during the pandemic, including those who work in the informal economy; citizen groups demanded the resignation of President López Obrador for his handling of the pandemic. Other forms of pandemic-related disorders include street vendors protesting closures of their businesses, residents protesting over unsubstantiated rumors of COVID-19 inducing substances being spread by drones, against checkpoints that limited access to certain cities, and over mask-wearing.

Similarly as to India, conflicts pre-dating the pandemic were paused in the early months of COVID-19 only to later intensify. Most notably, gang violence and drug cartel battles increased during the summer, as travel restrictions were lifted and the drug trade reorganized itself to adapt to the new circumstances [65]. Mexico has also witnessed a large increase in murders and especially femicides as COVID-19 restrictions have made women more vulnerable to domestic violence [66]. Despite the pandemic, demonstrations calling for the safety of women have continued. In Mexico 71% of the total population had internet access in January 2021 [67]. Fig 1 synthetizes the most important events in the unfolding of the pandemic in India, Israel and Mexico.

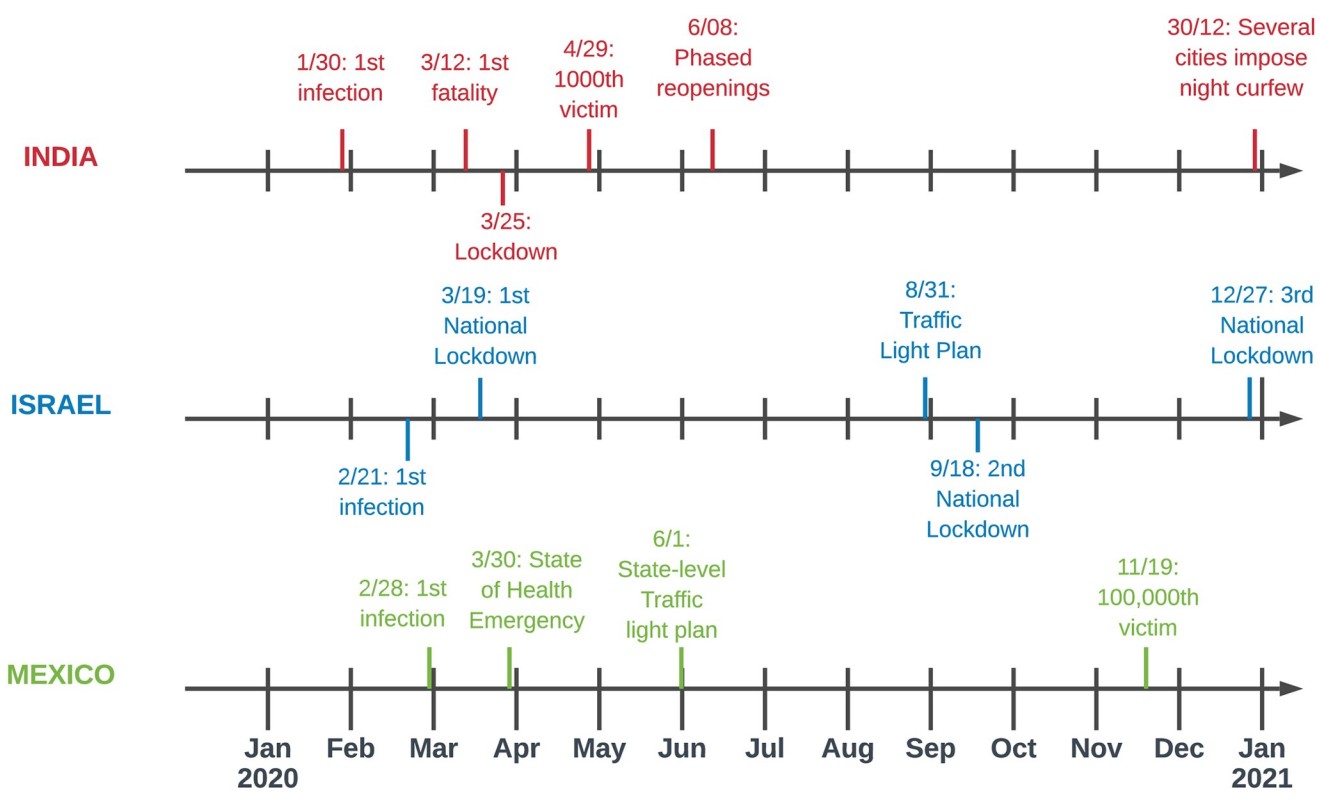

**Fig 1. Timeline of main COVID-19-related events in India, Israel and Mexico (January 2020-December 2020).**

## 3 Materials and methods

### 3.1 Data

Our analyses are based on data obtained from the COVID-19 Disorder Tracker (CDT) initiative of the Armed Conflict Location & Event Data Project (ACLED) [29]. ACLED is a well-known open-source data collection organization cataloguing global crises and conflicts, created to facilitate the study of political violence and social unrest. Its recent CDT initiative records events that are directly related to COVID-19 and excludes disorders that may temporally overlap with the pandemic but that are not directly related to it, such as conflicts between armed militias over a disputed territory (see [68]). The dataset includes, for example, protests against governments in response to COVID-19 decisions, attacks against COVID-19 healthcare workers, or against individuals who may allegedly spread the virus to others. Events are labeled as one of six types: violence against civilians, riots, protests, explosions, battles, and strategic developments. We exclude the strategic development category as it lists contextually relevant episodes that are not political violence but that may trigger, lessen or explain them. These are typically tactical changes by one of the relevant actors, such a government announcing a state of emergency or the easing of lockdown restrictions. In addition to type, the dataset records date and location of the event, the subjects involved, reported fatalities. Data is collected from governmental institutions, news media, humanitarian agencies, and research publications. The database is updated every week; in this work we included events occurring between January $3^{rd}$ to December $12^{th}$ 2020. Within this interval, a total of 20,135 disorder events were recorded worldwide. When considering weekly averages we will consider Sunday as the first day of the week and Saturday as the last. Hence, weekly averages will be conducted over the period January $5^{th}$ to December $12^{th}$ 2020, yielding

**Table 1. Distribution of disorder events (violence against civilians, riots, protests, battles) across the top 10 countries as tallied by ACLED between January 3<sup>rd</sup> and December 12<sup>th</sup> 2020.** The countries considered in this study, India, Israel and Mexico are italicized, and account for almost 40% of the world total. At this time of writing, ACLED does not report data for the US.

| Country | Recorded Events | % |
|---|---|---|
| *Israel* | *3,531* | *18.2* |
| *India* | *2,910* | *15.0* |
| *Mexico* | *1,276* | *6.6* |
| Argentina | 988 | 5.1 |
| Brazil | 920 | 4.7 |
| Pakistan | 588 | 3.0 |
| South Korea | 525 | 2.7 |
| Chile | 466 | 2.4 |
| Peru | 460 | 2.4 |
| Morocco | 429 | 2.2 |
| Other (128 countries) | 7,318 | 37.7 |

weeks 2 to 50 of the year 2020. Week 1 of 2020 is the period between December 29<sup>th</sup> 2019 and January 4<sup>th</sup> 2020. Since disorder events were first recorded on January 3<sup>rd</sup> 2020 we assume all prior dates in week 1 carry zero events.

Table 1 lists the ten countries with the highest number of incidents; we focus on the first three: India, Israel and Mexico. These countries alone account for almost 40% of all the events included in the CDT dataset. Fig 2 shows the distribution of event type per country. As can be seen, most are protests, with a smaller number of riots (mostly in India and to a lesser degree in Mexico), a marginal presence of violence against civilians, and an almost negligible number of battles. No explosions were recorded in any of the countries of interest, leaving only four relevant categories as detailed in the S1.1 section in S1 File.

Fig 3 displays the time-series of all types of events for the countries we investigate. As can be seen, trends differ due to the different political timelines. In India, the first pandemic-related event was recorded in January 2020, but was followed by only five others over the first two months of the year. Most disorders in India occurred in April (431 events, 14.8% of the countrywide total), May (592 events, 20.3% of the countrywide total), and June (481 events, 16.5% of the countrywide total). On June 29<sup>th</sup> 123 protests were tallied nationwide, the largest number during the period of interest and corresponding to 4.2% of the total number of events in India. These protests were promoted by Congress in opposition to increasing fuel prices despite economic hardships due to the coronavirus crisis. In Israel disorders first emerged in March 2020, however the greatest number of demonstrations occurred in October (1,427 events, 40.4% of the countrywide total) and in November (1,037 events 29.4% of the countrywide total). The dramatic spikes observed in this period overlap with the lockdown orders announced at the onset of the second wave of infections; the Flag Movement promoted nationwide demonstrations. In Mexico, disorders also first appeared in March 2020. However, the distribution of events here seems to be more uniform than in India and Israel; this may be due to Mexico never having imposed a complete, countrywide lockdown. The highest recorded number of disorders in the country were tallied on March 30<sup>th</sup> (36 events) when health workers protested against the lack of medical supplies and equipment, and on April 27<sup>th</sup> (28 events) and May 25<sup>th</sup> (32 events) when citizens demanded better economic and financial aid. Overall, the most demonstrations took place in April (289 events, 22.7% of the countrywide total) and in May (275 events 21.6% of the countrywide total).

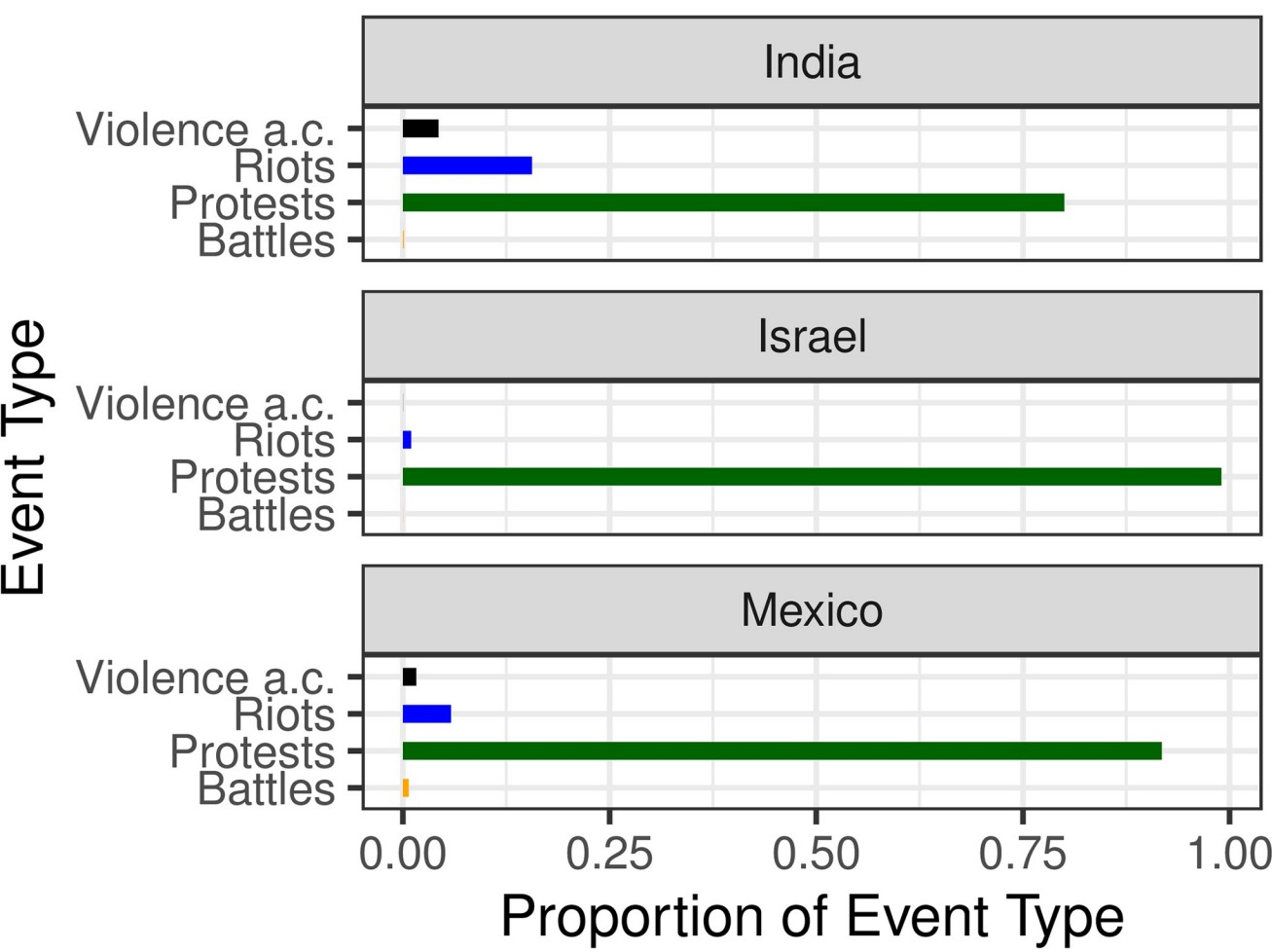

**Fig 2. Event type distribution across India, Israel and Mexico as tallied from January 3rd to December 12th, 2020.** "Violence a.c." stands for "Violence against civilians". The majority of events are protests, followed in smaller percentage by riots, violence against civilians, and battles respectively. ACLED also lists explosions, which are not reported in any of the countries under investigation.

**3.1.1 Spatial clustering.** Spatial-temporal concentration is a well-known signature of social phenomena, including disorder events [50, 69]. To examine the spatial distribution of disorders as listed in the CDT dataset for the three countries of interest, we first identify subnational geographic areas where events colocalize. We do this by using $k$-means clustering, a well-known algorithm which assigns each event to one of $k$ clusters by iteratively updating the centers of these clusters and minimizing the root-mean-square distance between the event location and its assigned cluster center [70]. The number of clusters $k$ is a parameter for the algorithm; as it increases, the average mean distance of events from their assigned center typically decreases. However, beyond a critical threshold $k^*$, the decrease rate may become negligible, indicating that new clusters are not distinguishable from old ones. As shown in S1.2 section in S1 File, applying $k$-means clustering to the CDT data for India, Israel and Mexico, yields $k^* = 4$ as this critical threshold for all three. Henceforth, we will consider four geographical clusters per country.

**3.1.2 Hawkes process.** The temporal Hawkes process [32, 71, 72] models the distribution of a random time variable as a non-homogeneous Poisson process with a time-dependent

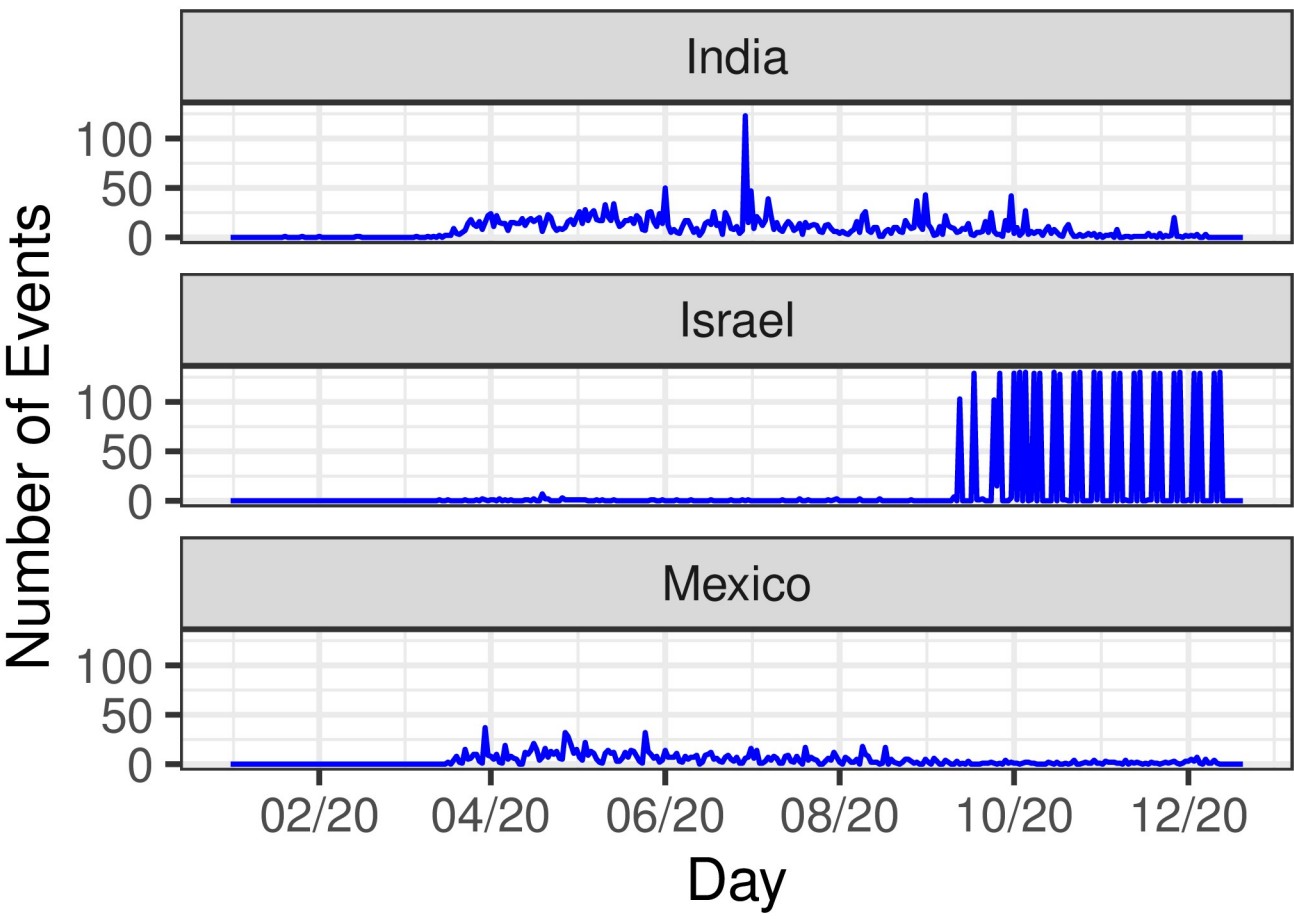

**Fig 3. Time series of nationwide disorder events (protests, riots, violence against civilians, battles) in India, Israel and Mexico, from January 3$^{rd}$ to December 12$^{th}$ 2020.** Note the relatively more uniform distribution in Mexico, compared to the more structured ones in India and especially in Israel.

intensity function $\lambda(t)$ defined as

$$\lambda(t) = \lim_{h \to 0} \frac{\mathbb{E}(N(t+h) - N(t))}{h}. \tag{1}$$

Here $N(t) = \sum_i \mathbf{1}_{t_i \leq t}$ is number of events up to time $t$. The index $i$ labels the events so that the event time sequence $\{t_i\}$ defines the point process. Thus, $\lambda(t)$ measures the number of events that are expected to arrive per unit time. For a homogeneous Poisson process $E(N(t)) = \lambda t$ is proportional to time and $\lambda(t) = \lambda$ is constant. Within non-homogeneous Hawkes processes $\lambda(t)$ is typically decomposed into a background intensity $\mu(t)$, often assumed to be constant $\mu(t) = \mu$, and an excitatory component $g(t)$ triggered at the times $t_i$ of past events so that

$$\lambda(t) = \mu + \sum_{t_i < t} g(t - t_i) \tag{2}$$

We model the excitatory function $g(t)$ through an exponential decay, according to standard protocols

$$\lambda(t) = \mu + \sum_{t_i < t} \alpha e^{-\beta(t - t_i)}, \tag{3}$$

where $\alpha$ and $\beta$ quantify the self-excitatory degree of the process. Here, $\alpha$ represents a jump factor representing the rate of increase of events immediately after a triggering, prior event, while $\beta$ is the associated decay rate; $1/\beta$ is often used as a proxy for the typical lifetime of an excitation. Large values of $\alpha$ and small values of $\beta$ imply the process is highly reactive and its effects last longer, respectively [20]. The $\alpha$ and $\beta$ parameters are learned by applying maximum likelihood estimation (MLE) to Eq 3 where $\{t_i\}$ is known and the $\mu, \beta, \alpha$ parameters are to be determined, as described in S1.3 Section in S1 File. From $\alpha$ and $\beta$ one can derive the branching ratio $\gamma$, an estimate of the total number of events that are endogenously generated by a single event

$$\gamma = \int_0^\infty \alpha e^{-\beta t} dt = \frac{\alpha}{\beta}. \tag{4}$$

If one considers "immigrant" events those that occur independently within a given generation, and the ones they trigger at the next generation as their "offspring", $\gamma$ may also be interpreted as the total expected number of offsprings triggered by an immigrant event. The so-called supercritical regime $\gamma > 1$ implies that the number of offspring events is larger than the number of immigrant events that generated them, leading to the unrealistic scenario of an infinite cascade. Hence, the $\gamma < 1$ constraint is imposed in the maximum likelihood estimation. The sub-critical regime ensures that the cascade of events triggered by an original immigrant event will eventually subside. Thus, assuming $\gamma < 1$, we can also estimate the total number of offspring events $\mathcal{N}_\infty$ generated by a single immigrant event. If at each generation $\gamma$ offspring events arise, the number of events at generation $j$ is given by $\gamma^{j-1}$, where $j = 1$ represents the first, single, immigrant event. Hence the total number of offspring events is

$$\mathcal{N}_\infty = \sum_{j=1}^\infty \gamma^{j-1} = \frac{1}{1-\gamma}. \tag{5}$$

Using Eq 5 we can estimate the average number of events until time $t$, due to both background and excitatory events

$$E(N(t)) \approx \frac{\mu t}{1-\gamma}, \tag{6}$$

from which the average expected number of events per unit time can be evaluated

$$\lim_{t \to \infty} \frac{E(N(t))}{t} \approx \frac{\mu}{1-\gamma} = \mu\left(1 + \frac{\gamma}{1-\gamma}\right). \tag{7}$$

The last equality in Eq 7 implies that $\gamma < 1$ also represents the percent of events per unit time that are endogenously generated. We apply Eq 2 to our data, both as a baseline Poisson process (setting the self-excitability function $g = 0$) so that $\lambda(t) = \lambda = \mu$ and making the process Markovian, and as a Hawkes process ($g \neq 0$, and as in Eq 3). To compare results, we determine the Akaike Information Criterion (AIC) values of the two models defined as

$$\text{AIC} = 2\kappa - 2\log L \tag{8}$$

where $\kappa$ is the total number of parameters used ($\kappa = 1$ for a Poisson process and $\kappa = 3$ for a Hawkes process) and $\log L$ is the MLE of the model as described in S1.3 section in S1 File. By construction, the model with the lowest AIC value is the one that best fits the data.

We also employ residual analysis to validate the choice of the exponential function $g$ in modeling the self-excitability of the process [73] Given the intensity function $\lambda(t)$ of a Hawkes

process and the set of event times $\{t_i\}$ we can derive the set of residuals $\{\tau_i\}$ defined as

$$\tau_i = \int_0^{t_i} \lambda(t) \mathrm{d}t \tag{9}$$

It can be shown that the $\{\tau_i\}$ residuals are independent and follow a stationary Poisson process with unit rate [73, 74]. This implies that the inter-arrival values $Y_i = \tau_i - \tau_{i-1}$, defined for $i > 1$ and where $\tau_0 = 0$ is imposed, define a set of independent and exponentially distributed variables. Finally, it follows that the derived quantities $U_i = 1 - \mathrm{e}^{-Y_i}$ are also independent and uniformly distributed. To test the goodness of fit of the Hawkes process we can thus verify whether the $0 \le U_i < 1$ values are indeed uniformly distributed. Operationally, we employ the two-tailed Kolmogorov-Smirnov (KS) test, a common non-parametric test that determines whether a set of given observations (in this case $\{U_i\}$) come from a known distribution (in this case the uniform distribution between 0 and 1) [75]. The test compares the value of the statistic $D$

$$D = \max\left[ \left( \max_i \left| U_i - \frac{i-1}{\mathcal{U}} \right| \right), \left( \max_i \left| U_i - \frac{i}{\mathcal{U}} \right| \right) \right], \tag{10}$$

to a given critical value $D_c$. In Eq 10, $\mathcal{U}$ is the cardinality of the $\{U_i\}$ set, which is the number of observations and same as the cardinality of the $\{Y_i\}$ and $\{\tau_i\}$ sets. If $D > D_c$, then the hypothesis that the $\{U_i\}$ values follow a uniform distribution, and hence that the $\{t_i\}$ values define a Hawkes-like point process, can be rejected at the $\alpha$ level of significance. We apply the KS test to our data at the 95% $D_\alpha = 1.36/\sqrt{\mathcal{U}}$, $\alpha = 0.05$) and at the 99% ($D_\alpha = 1.36/\sqrt{\mathcal{U}}$), $\alpha = 0.01$) confidence levels. Other tabulated values for $D_c$ and different confidence levels can be used [76].

## 4 Results

### 4.1 India

The four spatial clusters we identified in India are visualized in Fig 4; for the most part they follow geographical and/or topographical divisions within the country. Cluster 1 (C1), which includes the northern states of Himachal Pradesh, Rajasthan, Haryana, Uttar Pradesh, Uttarkhand, and the Union territories of Jammu and Kashmir and Ladakh, accounts for a total of 913 events. Cluster 2 (C2) is associated with the highest number of events overall, 946, and covers the eastern states of Arunachal Pradesh, Assam, Bihar, Jharkhand, and Odisha. The third cluster (C3) groups 436 in the central and western states of Maharashtra, Gujarat, and Madhya Pradesh. Finally, Cluster 4 (C4) accounts for a total of 568 events in the southern states, including Kerala, Tamil Nadu, Andhra Pradesh, Karnataka, and Telangana.

Fig 5 displays the number of events $n_j$ occurring during week $j$ in each of the C1-C4 clusters. For all clusters we observe the first spiking of events around week $j = 13$ and $j = 14$ (between March 22nd and April 4th 2020) followed by a second period of intensifying activity around week $j = 27$ (between June 28th and July 4th), although the relative magnitude of events are cluster-dependent. Cluster-wise event distributions for India are also visualized in box plots in S1.4.1 section in S1 File.

The left panel of Fig 6 displays Pearson's correlation coefficient $r$ for the weekly number of events $\{n_j\}$ between pairs of clusters. This quantity ranges from $r = 0.595$ between C2 and C3 to $r = 0.724$ between C1 and C3, showing relative synchrony among clusters. Starting from $\{n_j\}$ we can also construct the first-order difference sequence $\{\Delta n_j\}$ where $\Delta n_j = n_j - n_{j-1}$, and investigate how differences in weekly counts correlate between clusters. The right panel of Fig 6 shows that when using $\{\Delta n_j\}$ the positive association between clusters lowers significantly

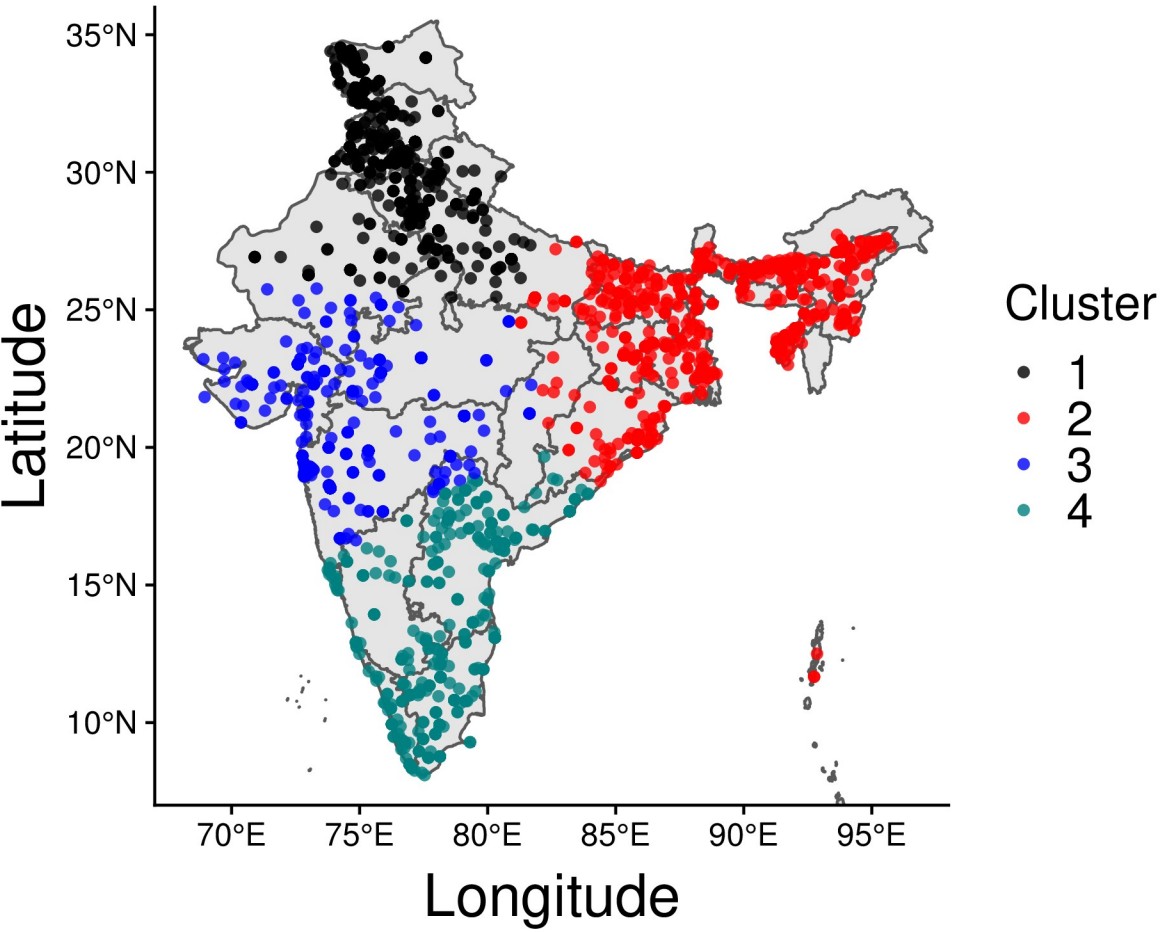

**Fig 4. COVID-19 disorder events in India.** The four detected clusters, C1-C4, host approximately equal numbers of residents, with a maximum of about 27% of the total population in C1 and a minimum of about 18% in C2. Clusters however are very heterogeneous in terms of population density and territorial extent. Two of the most densely inhabited states in the area, Uttar Pradesh and Bihar, are located in C1 and C2, respectively. This map has been generated via rnaturalearth in R, a package built using Natural Earth map data.

compared to when $\{n_j\}$ is used, but remains significant in some clusters indicating partial national synchrony. Specifically, C3 and C4 carry the highest correlation value, $r = 0.512$, whereas the weakest relationship is between C1 and C2, for which $r = 0.124$. Noteworthy is the relationship between C1 and C3, which has the highest $r$ relative to $\{n_j\}$ counts and the second-lowest $r$ relative to $\{\Delta n_j\}$ counts. Numerical values are listed in S1.5.1 subsection in S1 File.

Finally, in Table 2 we list statistical outcomes pertaining to the India disorders, where we tally events daily. Here, the Hawkes process with an exponential excitatory term $g$ always leads to lower AIC values compared to the baseline Poisson process, indicating a considerable degree of self-excitability and temporal dependence in the data. All cases considered (national and subnational Hawkes processes) passed the KS test, as can be verified by the $D$ values being lower than the critical KS value in Table 2. We tally 2,910 countrywide disorders, which if treated as a Hawkes process on the national scale, lead to a background rate of $\mu = 0.5$ events per day with self-excitatory events arising at a rate of $\alpha = 2.07$ events per day, spanning over $1/\beta = 0.46$ days. These values lead to a branching ratio $\gamma = \alpha/\beta = 0.95$, revealing that a large percentage of disorder events are due to feedback mechanisms.

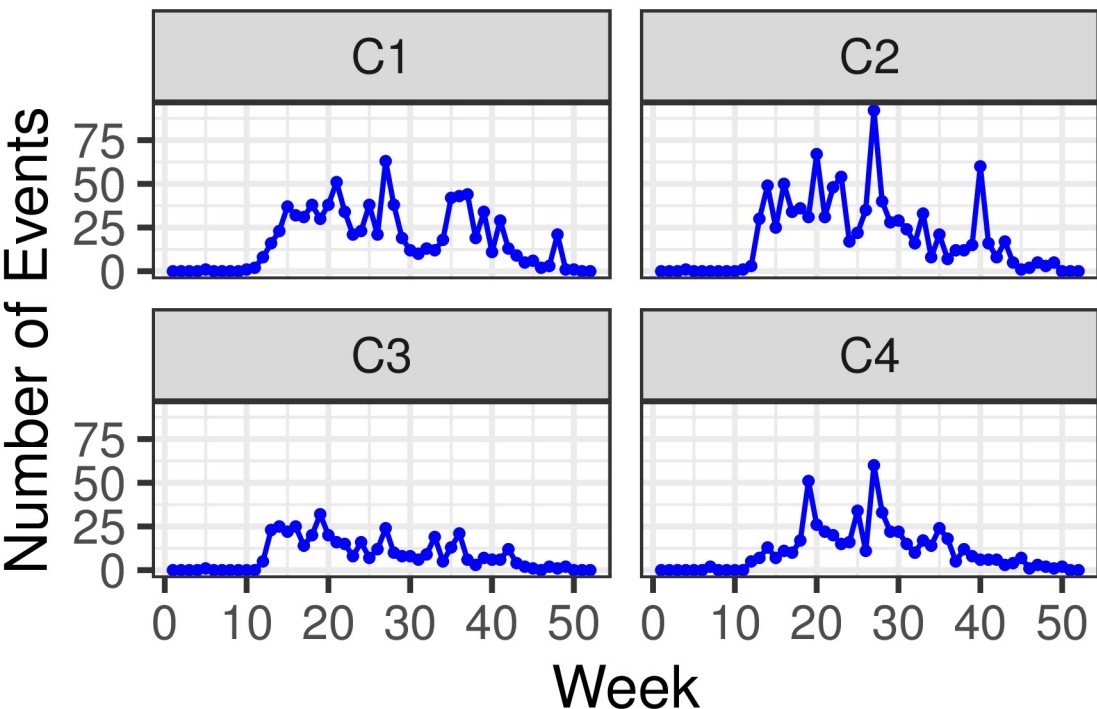

**Fig 5. Weekly time series of disorder events $\{n_j\}$ visualized by cluster, in India.** Weeks are marked from week $j = 1$ (December 29th 2019 to January 4th 2020) to week $j = 50$ (December 6th to December 12th 2020).

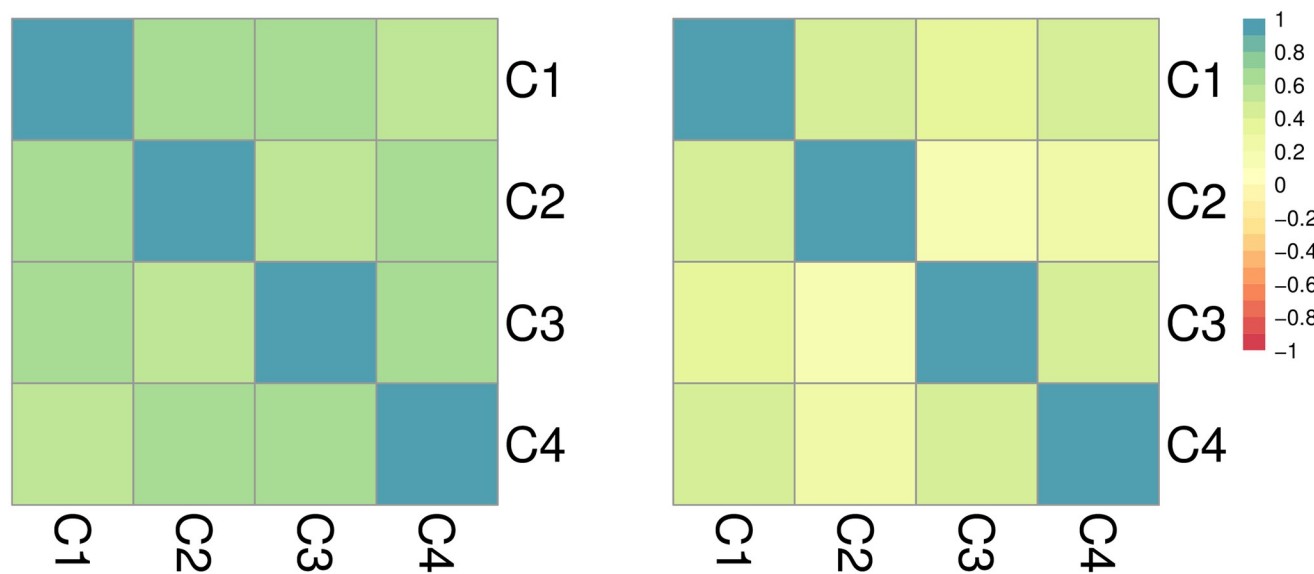

**Fig 6. Cluster dynamics in India: (left) Pearson's correlation of weekly events** $n_j$ **across pairs of clusters.** (right) Pearson's correlation of differentiated weekly events $\Delta n_j = n_j - n_{j-1}$. The color scale is restricted to positive values as as no negative relationships are found. The left panel reveals a substantial level of correlation, however, when first-order differences are considered all coefficients decrease, implying that the rate of change in the occurrence of events is less correlated.

**Table 2. Statistical outcomes of the Hawkes process applied to data from India.** The Hawkes process outperforms the baseline Poisson process both nationwide and in each cluster, since the Hawkes AIC is always less than the Poisson AIC. The Hawkes process passes the KS test at the 95% significance level in all cases, with $D < D_c^{95}$.

| Cluster | India (all) | India (C1) | India (C2) | India (C3) | India (C4) |
|---|---|---|---|---|---|
| Number of events | 2,910 | 913 | 993 | 436 | 568 |
| $\mu$ | 0.497 | 0.611 | 0.368 | 0.173 | 0.322 |
| $\alpha$ | 2.073 | 1.529 | 1.400 | 0.569 | 0.712 |
| $\beta$ | 2.192 | 1.928 | 1.586 | 0.646 | 0.854 |
| $\gamma$ | 0.946 | 0.793 | 0.882 | 0.881 | 0.834 |
| $\mu/(1-\gamma)$ | 9.15 | 2.95 | 3.13 | 1.44 | 1.93 |
| Hawkes AIC | -9298 | -768 | -1178 | 286 | 23 |
| Poisson AIC | -6983 | -144 | -261 | 579 | 403 |
| KS Stat, $D$ | 0.049 | 0.051 | 0.084 | 0.125 | 0.063 |
| KS Crit 95%, $D_c^{95}$ | 0.112 | 0.098 | 0.127 | 0.205 | 0.142 |
| KS Crit 99%, $D_c^{99}$ | 0.135 | 0.118 | 0.152 | 0.245 | 0.170 |

On the more local level, C1 and C2, the clusters with the largest number of events, display similar self-excitatory trends compared to C3 and C4. As can be seen from Table 2 values of $\alpha$ are larger in C1 and C2 ($\alpha$ = 1.52 and 1.40 events per day, respectively) than in C3 and C4 ($\alpha$ = 0.57 and 0.71 events per day, respectively) however the associated timescales $1/\beta$ are less than one day in C1 and C2 ($1/\beta$ = 0.5 and 0.6 days, respectively), smaller compared to those observed for C3 and C4 ($1/\beta$ = 1.54 and 1.2 days, respectively). These results suggest that C1 and C2 are marked by more intense, yet more quickly damped feedback activity than C3 and C4. The two opposite trends lead to relatively uniform branching ratios across clusters with $\gamma$ ranging from $\gamma$ = 0.88 (C2) to $\gamma$ = 0.79 (C1). Of particular interest is C3, the most sparse cluster, as can be seen in Fig 4). This cluster also carries the least number of events (436) and displays the lowest reactivity yet, it displays the longest timescale, leading to a very large branching ratio $\gamma$ = 0.88. These results suggest that although disorder events are rarer in C3, feedback effects are very strong and echoes of disorder persist the longest. Finally, the background intensity $\mu$ is highest in C1 ($\mu$ = 0.61 events per day) and smallest in C3 ($\mu$ = 0.17 events per day). Overall, one may expect the emergence of roughly 9 events per day countrywide, with a high likelihood of events being self-excited; disaggregating trends within the separate clusters shows that most of these events are to be expected in C2 and C1, and to a lesser degree in C4 and C3. While C3 contributes less than others to the expected daily disorder count, the degree of self-excitation is strong. Interestingly, C1 and C2 are also the clusters that according to the 2011 Census of India, host the states with the most dense population: Uttar Pradesh in C1 (200 million residents, and a density of 828 persons per km$^2$) and Bihar in C3 (105 million residents, and a density of 1,102 persons per km$^2$).

## 4.2 Israel

We apply similar procedures for disorder events in Israel, leading to the four clusters displayed in Fig 7. Cluster 1 (C1) groups 792 events occurring in the greater Jerusalem area and in proximity of the Gaza strip; the second cluster (C2) is located in the southern part of the country, at the border with Jordan and only counts 220 events. The third cluster (C3) is centered around Tel Aviv, on the western coast and contains 1073 events. Finally, cluster C4 is located in the Haifa region, in the northern part of the country, and contains the most number of events, 1446.

Fig 8 shows the temporal dynamics across the four clusters in Israel. From a temporal distribution standpoint, the overall picture is instead sensibly different from what observed in India.

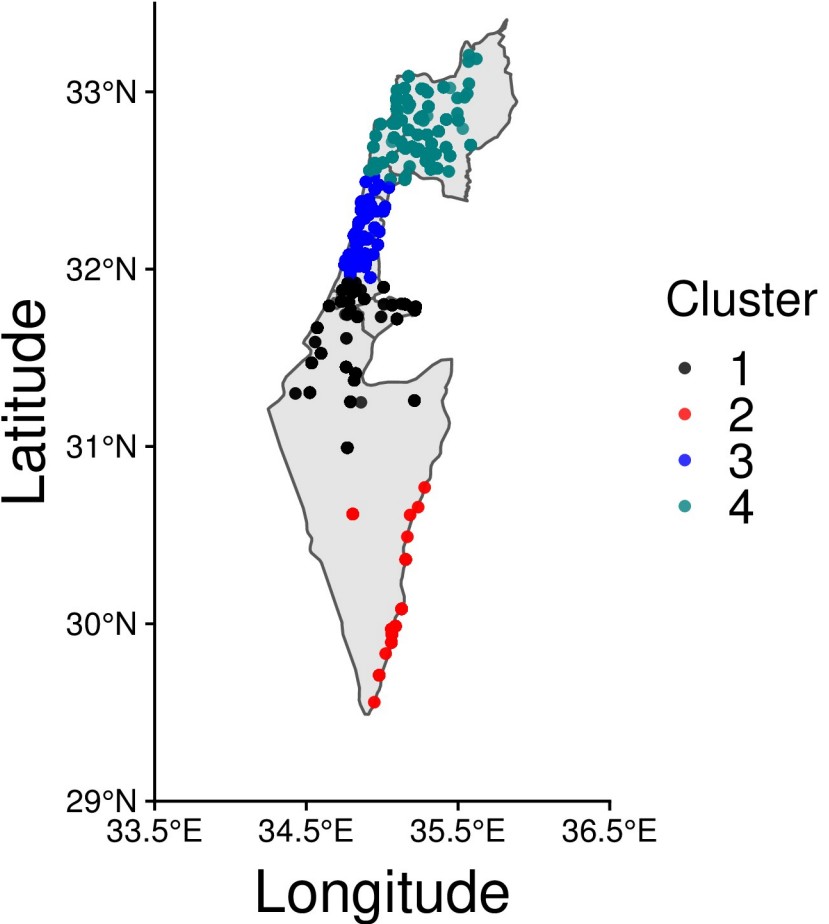

**Fig 7. COVID-19 disorder events in Israel.** Clusters C1, C3, C4 host the most densely populated areas located around the cities of Haifa, Tel Aviv and Jerusalem, respectively, Events in cluster C2, the least dense region, are the most sparse and emerge mostly at the border with Jordan. This map has been generated via rnaturalearth in R, a package built using Natural Earth map data.

Besides scattered events recorded in early 2020, most weekly events $n_j$ are concentrated between week $j = 37$ (September 6[th] to September 12[th]) and week $j = 50$ (December 6[th] to December 12[th]), concurrent with nationwide protests organized by the Flag Movement against the alleged corruption of Prime Minister Nethanyahu and his failures in managing the pandemic Cluster-wise event distributions for Israel are also visualized in box plots in S1.4.2 section in S1 File.

The Pearson's correlation coefficients $r$ comparing weekly $\{n_j\}$ events between pairs of clusters are shown in Fig 9 and reveal strong homogeneous, positive synchrony Values range between $r = 0.995$ (between C2 and C3, and C2 and C4) and $r = 0.999$ (between C3 and C4). Similarly strong, positive relationships persist when considering the first order difference sequence $\{\Delta n_j\}$. This finding underscores that even when considering increasing or decreasing trends, clusters are tightly aligned, indicating nation-wide synchrony. The fact that this feature emerges with so much clarity in Israel compared to India may be a consequence of its much smaller territorial extent, a more linguistically homogenous population, and/or larger access to broadband internet, but also the result of the Flag Movement's nation-wide campaigns. Numerical values are listed in S1.5.2 subsection in S1 File.

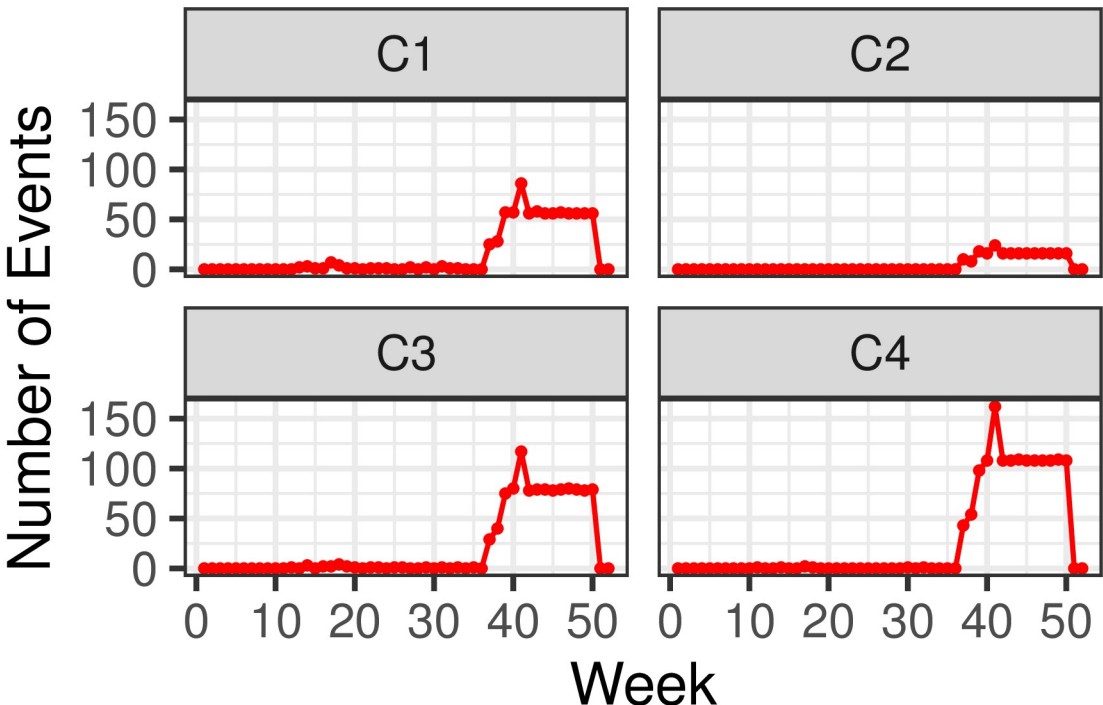

**Fig 8. Weekly time series of disorder events {$n_j$} visualized by cluster, in Israel.** Weeks are marked from week $j = 1$ (December 29th 2019 to January 4th 2020) to week $j = 50$ (December 6th to December 12th 2020).

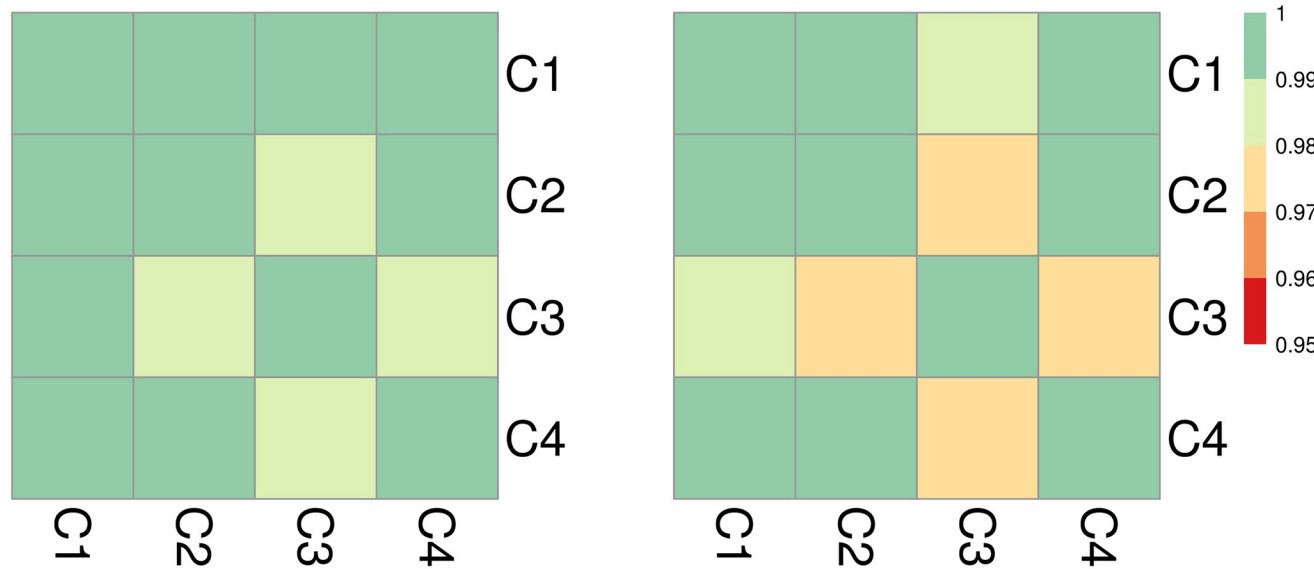

**Fig 9. Cluster dynamics in Israel: (left) Pearson's correlation of weekly events $n_j$ across pairs of clusters.** (right) Pearson's correlation of differentiated weekly events $\Delta n_j = n_j - n_{j-1}$ across pairs of clusters. The left panel shows almost perfect correlation between weekly-based streams of events for all cluster pairs. The synchrony remains almost perfect when considering first-order differences in the right. The nationwide correlation that is much more visible than in India or Mexico may be due to Israel's more compact geographical extension, linguistic unity, tighter virtual connectivity, and/or due to the nationwide engagement of the Flag movement. Note that correlation coefficients between clusters in Israel are very large ($r \geq 0.95$ in all cases) compared to those computed for India (and Mexico). Thus, if we kept the same scale as in Figs 6 and 12 ($-1 \leq r \leq 1$) the correlation plots for Israel would be colored uniformly. Instead, for a more nuanced view we use instead a more restricted scale ($0.95 \leq r \leq 1$).

**Table 3. Statistical outcomes of the Hawkes process applied to data from Israel.** The Hawkes process outperforms the baseline Poisson process both nationwide and in each cluster, since the Hawkes AIC is always less than the Poisson AIC. The Hawkes process passes the KS test at the 95% significance level in all cases except for C4, where $D > D_c^{99}$, indicating that the hypothesis that the data can be fit to a Hawkes process with a decaying exponential should not be accepted. Given the nature of the data, a more steeply decaying function than the decaying exponential should be used in C4.

| Cluster | Israel (all) | Israel (C1) | Israel (C2) | Israel (C3) | Israel (C4) |
|---|---|---|---|---|---|
| Number of events | 3,531 | 792 | 220 | 1,073 | 1,446 |
| $\mu$ | 0.390 | 0.265 | 0.590 | 0.236 | 0.151 |
| $\alpha$ | 23.528 | 9.894 | 4.822 | 12.386 | 15.306 |
| $\beta$ | 24.212 | 10.812 | 6.310 | 13.118 | 15.717 |
| $\gamma$ | 0.972 | 0.915 | 0.764 | 0.944 | 0.974 |
| $\mu/(1-\gamma)$ | 13.79 | 3.12 | 2.50 | 4.21 | 5.79 |
| Hawkes AIC | -25047 | -2655 | -145 | -4588 | -7649† |
| Poisson AIC | 10988 | -149 | 58 | -805 | -1917 |
| KS Stat, $D$ | 0.098 | 0.157 | 0.134 | 0.156 | 0.274 |
| KS Crit 95%, $D_c^{95}$ | 0.127 | 0.160 | 0.190 | 0.168 | 0.212 |
| KS Crit 99%, $D_c^{99}$ | 0.153 | 0.192 | 0.228 | 0.202 | 0.254 |

Table 3 lists all statistical quantities derived from fitting the Hawkes process to the tabulated disorder events in Israel. As observed for India, the Hawkes process always outperforms the baseline Poisson process in terms of AIC values. However, the $D$ statistic derived from the KS analysis is higher than the critical $D_\alpha$ both at the 95% and at the 99% confidence levels, indicating that more appropriate time-dependent point processes should be used to describe the data. For example, the extreme clustering that characterizes events in C4 might be better represented by more rapidly decaying forms than the exponential decay $g(t)$. From Table 3 we also observe that the 3,531 countrywide events can be described by a Hawkes process with a decaying exponential function $g$ where $\alpha = 23.53$ events per day, lasting $1/\beta = 0.45$ days. These values correspond to a branching ratio $\gamma = 0.97$ and indicate that disorders cause strong feedback of short duration. This is confirmed by Fig 3 where we observe a very high concentration of events occurring over a limited timeframe. Clusters C1, C3 and C4 display relatively similar trends: the background rate $\mu$ varies between $\mu = 0.151$ (C4) and $\mu = 0.265$ (C1) events per day, whereas $\gamma$ ranges from $\gamma = 0.92$ (C1) to $\gamma = 0.97$ (C4), indicating relatively low background rates but sustained feedback. Conversely, C2 reports the largest background rate among all clusters with $\mu = 0.59$ events per day, the lowest reactivity with $\alpha = 4.82$ events per day and the largest lifetime $1/\beta = 0.15$ days, which combine to yield the lowest branching ratio $\gamma = 0.76$ of all clusters. On average, the total number of disorders expected in Israel is $\mu/(1-\gamma) = 13.8$ events per day, arising in descending order in C3, C1 and C2. However, the contribution of C4 cannot be determined since the Hawkes process may not be the most adequate representation of the local point process distribution in this cluster.

## 4.3 Mexico

As for India and Israel, the clustering process yields four clusters in Mexico; they are shown in Fig 10. Two of them exhibit a sparse nature, with events distributed in a less dense manner compared to the other two. The first cluster (C1) only contains 139 events that however are spatially concentrated. This cluster covers the southern part of the country, and includes the states of Chiapas, Veracruz, Campeche, Yucatan, and Quintana Roo. The second cluster (C2) contains a total of 270 events, mainly located in the states of Durango, Tamaulipas, Nuevo Leon, and Nayarit. Cluster C3 is the least populated, tallying only 101 events that are spread across the Northern states, particularly Baja California Sur, Baja California, Sonora, Sinaloa,

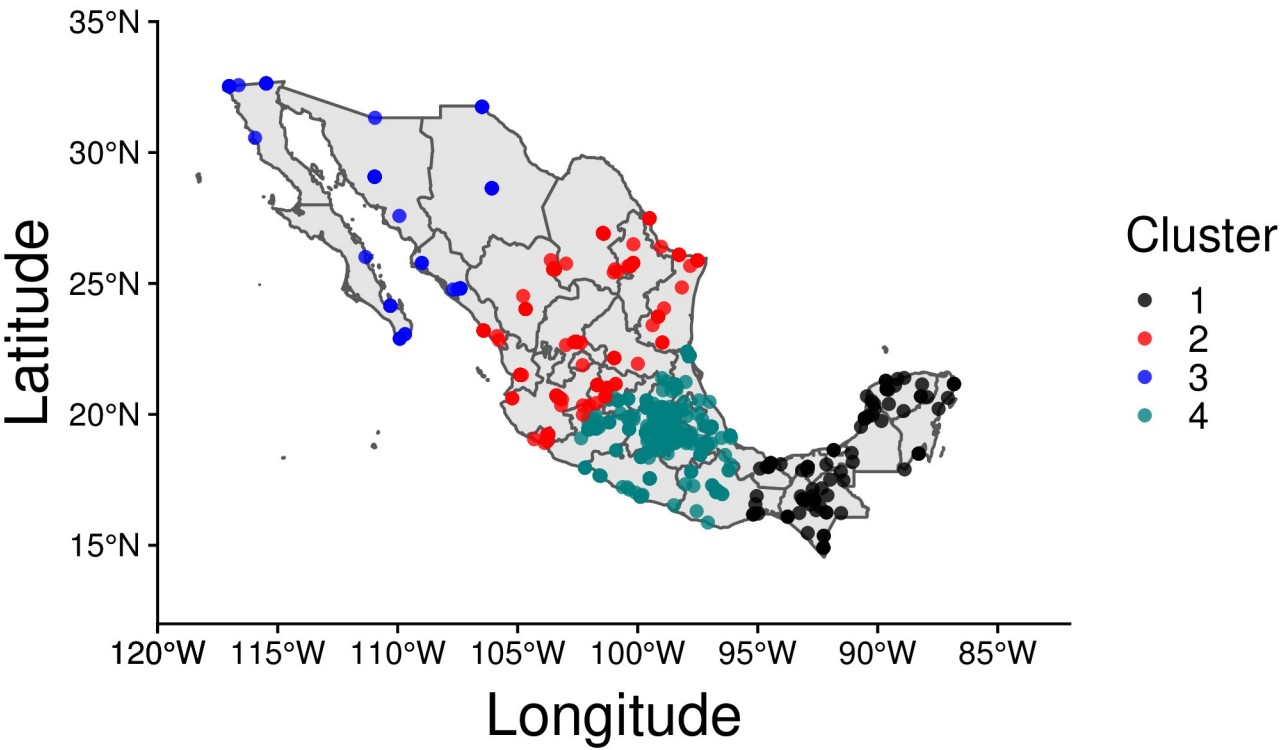

**Fig 10. COVID-19 disorder events in Mexico.** Of all clusters, C4 carries the largest population as it includes the capital city and the state of Mexico. The two are respectively the most populated city and state in the country. The state of Mexico is also the most dense nationwide. This map has been generated via rnaturalearth in R, a package built using Natural Earth map data.

and Chihuahua. As can be seen, the vast majority of events (766) are located in C4, which accounts for 60% of total disorders. C4 contains Mexico City, the country's capital, as well as the states of Oaxaca, Puebla, Queretaro, Guanajuato, Michoacan, and Mexico state.

Fig 11 displays cluster trends at the weekly level. Contrary to what observed for India and Israel, disorder events have mostly persisted throughout the COVID-19 crisis: in Mexico there have been no major interruptions of protests, riots or violence against civilians since the onset of the pandemic. This may be due to the government never having imposed a complete lockdown in the country. However, the alignment in peaks of disorder activity across clusters is weak, and much less pronounced than in India or Israel. For instance, while C4 (the cluster that records the majority of events) shows sustained activity on weeks $j = 14$ (March 29th to April 4th), $j = 16$ (April 12th to April 18th), and $j = 18$ (April 26th to May 2nd), similar trends are not detected in the other clusters. For example, in neighboring C2, activity spikes in week $j = 19$ (May 3rd to May 9th). Cluster-wise event distributions for Mexico are also visualized in box plots in S1.4.3 section in S1 File.

The Pearson's correlation coefficients $r$ for Mexico and its four clusters are shown in Fig 12. Calculations relative to the weekly $\{n_j\}$ events reveal some alignment between clusters, specially between C2 and C4 ($r = 0.826$), and to a lesser degree between C3 and C4 ($r = 0.772$). However, after computing the correlation coefficient relative to the first order difference sequence $\{\Delta n_j\}$, these relationships become much weaker: all positive correlations between weekly counts are reduced, C3 and C4 display low correlation ($r = 0.286$) and the value of the coefficients between C1 and C2 ($r = -0.092$) and C2 and C3 ($r = -0.445$) become negative. Numerical values are listed in S1.5.3 subsection in S1 File.

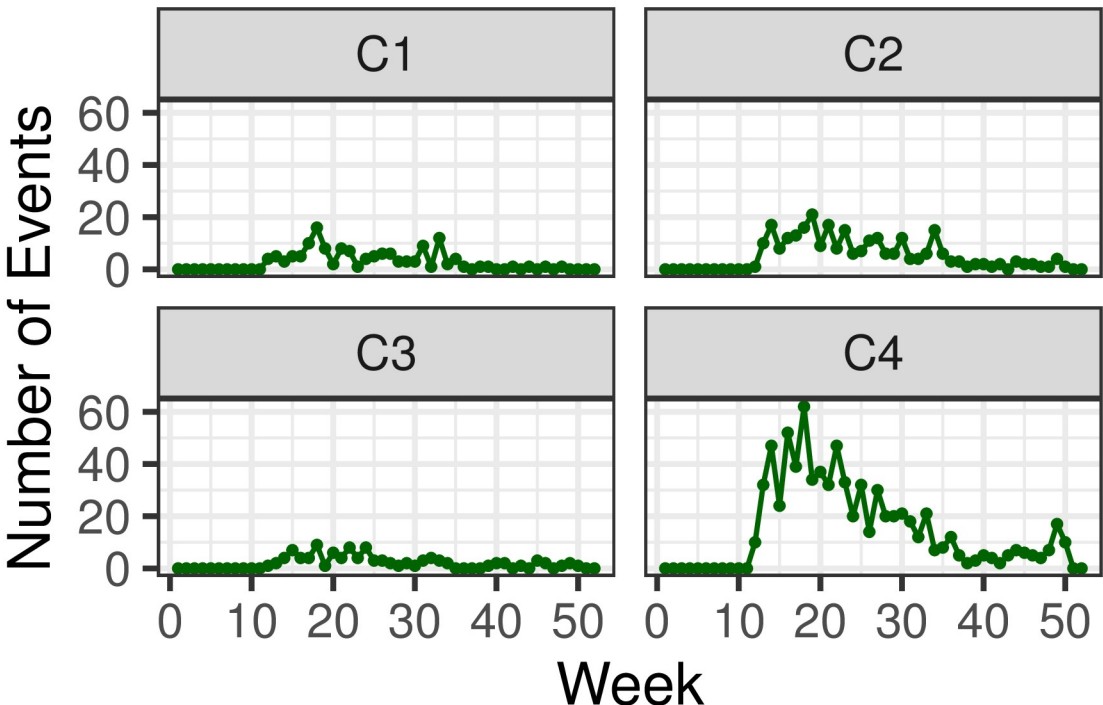

**Fig 11. Weekly time series of disorder events $\{n_j\}$ visualized by cluster, in Mexico.** Weeks are marked from week $j = 1$ (December 29th 2019 to January 4th 2020) to week $j = 50$ (December 6th to December 12th 2020).

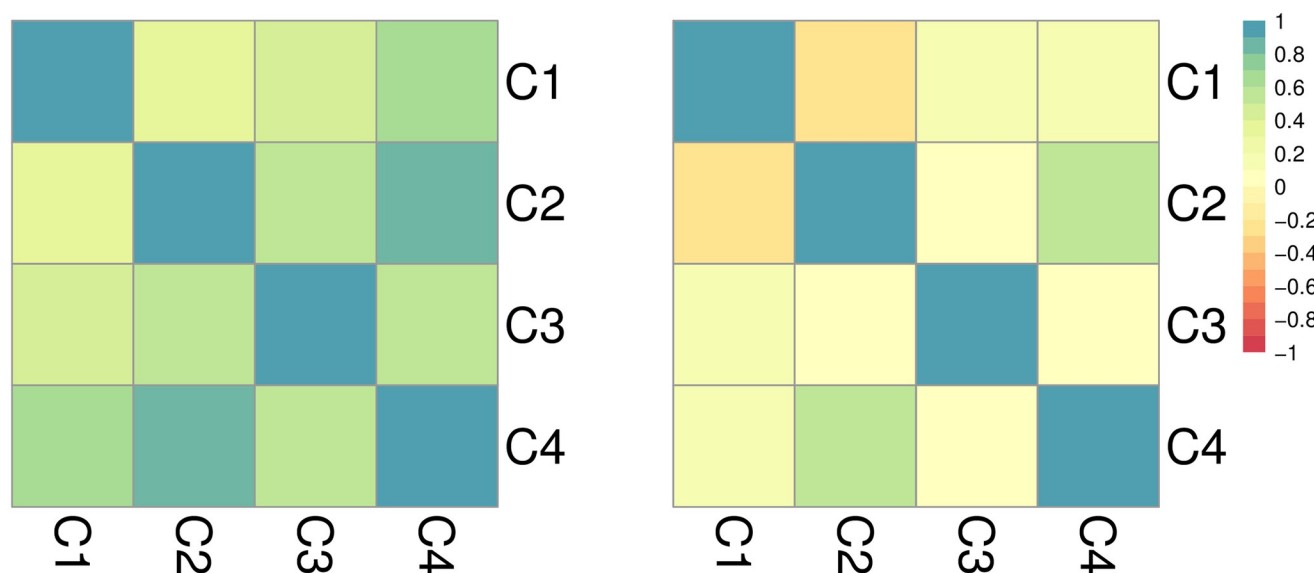

**Fig 12. Cluster dynamics in Mexico: (left) Pearson's correlation of weekly events $n_j$ across pairs of clusters.** (right) Pearson's correlation of differentiated weekly events $\Delta n_j = n_j - n_{j-1}$. The left panel shows a high level of correlation between clusters; however, contrary to what observed in India and Israel, dramatic decreases are observed when computing coefficients between first order differences implying a low level of synchrony in the rate of change of events.

**Table 4. Statistical outcomes of the Hawkes process applied to data from Mexico.** The Hawkes process outperforms the baseline Poisson process both nationwide and in each cluster, since the Hawkes AIC is always less than the Poisson AIC. The Hawkes process passes the KS test at the 95% significance level in all cases, with $D < D_c^{95}$.

| Cluster | Mexico (all) | Mexico (C1) | Mexico (C2) | Mexico (C3) | Mexico (C4) |
|---|---|---|---|---|---|
| Number of Events | 1,276 | 139 | 270 | 101 | 766 |
| $\mu$ | 0.996 | 0.217 | 0.469 | 0.121 | 0.774 |
| $\alpha$ | 1.931 | 0.446 | 1.086 | 0.101 | 1.660 |
| $\beta$ | 2.440 | 0.728 | 2.009 | 0.145 | 2.274 |
| $\gamma$ | 0.791 | 0.612 | 0.541 | 0.693 | 0.730 |
| $\mu/(1-\gamma)$ | 4.77 | 0.55 | 1.02 | 0.39 | 2.86 |
| Hawkes AIC | -2201 | 382 | 413 | 377 | -526 |
| Poisson AIC | -1424 | 445 | 532 | 401 | -69 |
| KS Stat, $D$ | 0.047 | 0.092 | 0.076 | 0.098 | 0.063 |
| KS Crit 95%, $D_c^{95}$ | 0.082 | 0.275 | 0.123 | 0.233 | 0.092 |
| KS Crit 99%, $D_c^{99}$ | 0.098 | 0.330 | 0.147 | 0.279 | 0.111 |

The statistical results obtained from applying the Hawkes model to Mexico are listed in Table 4. As for India and Israel, the Hawkes process yields better outcomes than the baseline Poisson process in modeling disorder events related to the COVID-19 pandemic, both nationwide and at the cluster level. Furthermore, all models fitted with data on Mexico passed the KS test as well, testifying to the goodness of fit provided by the specific Hawkes formalization of a self-excitability process. We identify an underlying temporal-dependence among events even in the case of C3, the cluster with least number of events. Countrywide, the 1, 276 events yield a Hawkes process marked by reactivity $\alpha$ = 2.29 events per day, with a relatively short lifetime of $1/\beta$ = 0.41 days. The branching ratio is $\gamma$ = 0.79, indicating a considerable amount of self-excitability, although much less than what observed in India and Israel. Among the various clusters, the greatest reactivity is observed in C4 with $\alpha$ = 1.66, decreasing in other clusters until a minimum of $\alpha$ = 0.10 events per day is reached in C3; C1 is also the cluster where the longest lifetime of self-excitatory phenomena is observed, with $1/\beta$ = 6.88 days whereas the shortest is in C4 where $1/\beta$ = 0.44 days. Overall, the cluster values of $\alpha$, $\beta$ yield sensibly lower values of $\gamma$ compared to India and Israel, ranging from $\gamma$ = 0.54 (C2) to $\gamma$ = 0.73 (C4) and confirming the nationwide trend. The background rate $\mu$ ranges from $\mu$ = 0.774 (C4) to $\mu$ = 0.121 (C3) events per day. Combined, these results imply that nationwide one can expect a total of $\mu/(1-\gamma)$ = 4.77 events per day, of which the most (2.86) will occur in C4, and the least (0.39) in C3.

## 5 Discussion and conclusion

We studied COVID-19 disorder events by using a public database compiled by the CDT initiative promoted by ACLED, the most reliable and complete source of data on conflicts and disorder patterns worldwide. Specifically, we analyzed the spatio-temporal distributions and characteristics of demonstrations in the three countries with the highest number of events *i.e.*, India, Israel, and Mexico between January 3rd and December 12th 2020. Using the well known Hawkes point process we investigated whether self-exciting effects could arise across events. We first considered countrywide data and later identified distinct geographical clusters in each of the three countries, to investigate trends on the more local level. Our intent was to better understand the macro- and meso-scale mechanisms that govern disorder events occurring in the same general context (the pandemic) but that may be ignited, shaped, acerbated or placated by more local happenings. Our work is in line with other empirical studies related to social

tension, nucleated by the seminal work of M. I. Midlarsky [27] and that include analyses of the 2011 London riots [40, 50], and of the 2005 Paris riots [48].

We identified four geographical clusters in each of the the countries we investigated by employing $k$-means clustering, These hosted varying numbers of events, were of varying spatial extent, and mostly followed clear geographical separations. We observed self-excitatory effects in all countries and in almost all subnational clusters, and found that the time-dependent Hawkes process is always a better fit to the disorder data than a simple Poisson process. We also performed several robustness checks, such as modifying the random number generator, or considering shorter time windows (see S1.6 section in S1 File); the number of clusters, and the applicability of the Hawkes process persisted in all cases. These results show that temporal dependence and self-excitability at the national level are not the result of the superimposition of unstructured, random processes at the subnational level. Instead, disorder events naturally cluster already at the local level, regardless of cluster size, and country examined.

However, while the temporal dependency between events, a hallmark of the Hawkes process, represents a common feature of protests, riots, and similar events related to the COVID-19 pandemic, important differences in the magnitude of these dependencies arise both intra- and inter-country. The three parameters that define the conditional intensity function of the Hawkes processes, $i.e.$ the background rate $\mu$, the reactivity $\alpha$, the decay rate $\beta$, as well as other derived quantities such as the branching ratio $\gamma$ and the average expected intensity $E[\lambda(t)]$, report a wide heterogeneity. This may be due to local infectivity trends, local decision-making, and how the pandemic and the associated measures impacted local calendars of religious or public holidays.

The distribution of disorder events in Mexico for instance, does not display large variations between March and October 2020; in India, although the frequency of disorders has always remained relatively large starting in April 2020, major spikes emerged in May 2020 and between June and July 2020. Israel, finally, displays an even more extreme situation: a handful of events were recorded in March 2020, but intense protesting occurred between September and October 2020. As a result of the relatively homogenous temporal trend, Mexico is associated with the largest background intensity $\mu$; reactivity $\alpha$ on the other hand is largest in Israel, the country which also displays the shortest duration of an excitation $\beta^{-1}$. In Israel, the influence of an event lasts for $\beta^{-1} = 0.04$ days, ten times less than in India and Mexico where $\beta^{-1} = 0.46$ and $\beta^{-1} = 0.41$ days, respectively, The more homogeneous course of events in Mexico is also manifest in the lowest branching factor, $\gamma = 0.791$, compared to India and Israel ($\gamma = 0.946$, $\gamma = 0.972$), implying that the probability that an event is endogenously generated as a consequence of another event is lowest in Mexico. Among the social and political events driving these patterns are the lack of a complete lockdown in Mexico, large scale protests in Israel as organized by the Flag Movement, the more compact geography and stronger internet connectivity in Israel compared to Mexico and India, which are also more linguistically and culturally heterogeneous. In all cases, the highest reactivity $\alpha$ emerges at the national scale, conversely, the average number of days upon which events may excite others is always highest in subnational clusters. Correlating the number of events with specific cluster demographic or geographical characteristics is outside the scope of this paper. However, regions with large numbers of pandemic-related events are often, but not always, characterized by large populations and/or large population densities. One notable exception is Jammu-Kashmir, a region in India marked by a relatively large number of events that is not among the country's most populous, nor most dense areas. Similarly, while the greater Jerusalem area is among the most densely inhabited in Israel, relatively few disorder events have been recorded here. These findings underly the need to consider the intersectionality of many demographic, socioeconomic,

political and/or religious factors when trying to understand why some clusters display more events than others.

Our work comes with some limitations. First, a more systematic approach to assess temporal dependence and self-excitability on the global scale is needed, especially considering the risk of prolonged restrictive measures during national vaccination campaigns. Although India, Israel, and Mexico report the highest number of disorders, they represent a small fraction of the countries in which disorders have occurred. Evaluating whether temporal clustering is a universal characteristic would help advance our knowledge on human behavior under prolonged periods of duress, and under greater social and cultural diversity. Furthermore, we did not distinguish events based upon their nature (*e.g.*, protests vs. violence against civilians), motives (*e.g.*, protests against restrictions imposed by the government vs. protests against the lack of supplies for health care workers; or participant type (migrant workers vs. students). As mentioned, the vast majority of events in India, Israel and Mexico, have been peaceful protests, however the dataset we utilized did not allow for a clear stratification of motives or participant type. Disentangling the distinct mechanisms that trigger self-excitability based on the qualitative features of the event themselves and the demands participants carry, would greatly add to our understanding of how social unrest unfolds.

Notwithstanding, our work shows that analyzing disorders at the national and subnational scales is useful. A national focus allows us to understand higher-level dynamics that transcend physical distances, especially in the current era of unprecedentedly fast information (and disinformation) spread. Social-media and other forms of long-distance connectivity can facilitate the dissemination of government decisions but also help large scale planning of nationwide or even international disorders. Understanding subnational dynamics allows for a more nuanced picture as local streams of protests may be ignited by more local issues. The global nature of the pandemic does not imply that its impacts are homogeneously distributed; to the contrary, existing socio-economic contexts lead to heterogeneous responses to the unfolding and management of the crisis. Decisions taken may resonate differently in different communities, which may be more or less concerned with the restriction of individual freedoms, being able to provide for one's livelihood, or prevent authoritarianism, and this may lead to localized waves of disorders, with a diverse composition of participants. Finally, our work underscores the need to align citizen trust with governmental decisions, especially at the local level, so that the implementation of restrictions and other public health measures are perceived to be temporary and in the interest of the common good, and that appropriate supporting policies are promoted to ensure the least economic disruption and uncertainty.

## Supporting information

**S1 File.**
(PDF)

## Author Contributions

**Conceptualization:** Gian Maria Campedelli, Maria R. D'Orsogna.

**Data curation:** Gian Maria Campedelli.

**Formal analysis:** Gian Maria Campedelli.

**Methodology:** Gian Maria Campedelli, Maria R. D'Orsogna.

**Software:** Gian Maria Campedelli.

**Supervision:** Maria R. D'Orsogna.

**Visualization:** Gian Maria Campedelli, Maria R. D'Orsogna.

**Writing – original draft:** Gian Maria Campedelli.

**Writing – review & editing:** Maria R. D'Orsogna.

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
