## [Decision Letter · Decision Letter 0]

17 Feb 2021

PONE-D-21-01644

Temporal Clustering of Disorder Events During the COVID-19 Pandemic

PLOS ONE

Dear Dr. Campedelli,

Thank you so much for submitting your manuscript to PLOS ONE. This round of review is now completed. You'll see that both authors feel positively about the manuscript, and have provided a number of quite detailed suggestions regarding various contextual, technical, and presentation issues. I encourage you to submit a revised version of the manuscript that addresses the points raised during the review process.

I look forward to receiving your revised manuscript!

Best,

Chad

Chad M. Topaz

Academic Editor

PLOS ONE

2. We note that Figures 3, 7 and 12 in your submission contain map/satellite images which may be copyrighted.

a. You may seek permission from the original copyright holder of Figures 3, 7 and 12 to publish the content specifically under the CC BY 4.0 license. 

Reviewers' comments:

Reviewer's Responses to Questions

**Comments to the Author**

1. Is the manuscript technically sound, and do the data support the conclusions?

Reviewer #1: Yes

Reviewer #2: Yes

2. Has the statistical analysis been performed appropriately and rigorously? 

Reviewer #1: Yes

Reviewer #2: Yes

3. Have the authors made all data underlying the findings in their manuscript fully available?

Reviewer #1: Yes

Reviewer #2: Yes

4. Is the manuscript presented in an intelligible fashion and written in standard English?

Reviewer #1: Yes

Reviewer #2: Yes

5. Review Comments to the Author

Reviewer #1: This manuscript presents an interesting application of Hawkes processes and k-means clustering to the spatial and temporal relationships between civil disorder events. In particular, the paper provides a quantitative evaluation of the suspected interplay between a government’s efforts to control the COVID-19 pandemic and resultant civil unrest. I do not have any major comments on the use of Hawkes processes for this data. I found the description of the method and communication of results very informative.

Major Comments:

Background

- Line 189+: When the BLM protests are discussed earlier in the paper, it’s stated that disorder such as this would not be included in the work since they “arise primarily in response to racial injustice, although racial disparities in COVID-19 cases may have contributed to the unrest.” This seems like a sensible exclusion, and I wonder if this exclusion criteria should also apply to the June CAA demonstrations. The authors seem to make the case that these are not primarily a result of the pandemic. In fact, they seem to be quelled rather than incited following lockdown orders.

Results

- I wonder if the box plots contribute enough to the overall objective to remain included as figures or if they could be removed in order to avoid detracting from the more informative results. There is very little main text dedicated to interpreting them in the Results section and none in the Discussion. Given the expected (and demonstrated) skew of the data, it seems like a less than ideal visualization technique. If the average count of weekly events is worth reporting explicitly, then that can be accomplished in the text and without a corresponding figure. Further, the distribution of weekly counts and its variability can be more clearly observed in the time series plots, which also allows the reader to see the temporal trends that lead up to "outlier" weeks.

Minor Comments:

Background

- Generally, these paragraphs are very dense with important and overlapping dates for each country. Perhaps a timeline figure would complement these descriptions and make this information easier to parse.

- Line 159-160: awkward sentence structure

- Line 165: “22nd”

- Line 168-169: awkward sentence structure

- Line 254: “including those who work in the informal economy”

Materials and Methods

- Fig. 1 caption:

o PLOS One guidelines require that all non-standard abbreviations be defined. Would recommend for “Violence a.c.”

o There’s a spelling issue in the database name (should be ACLED not ALCED)

o Consider including the date range for the data displayed to make the figure and caption stand alone.

Results

- Consider changing the x-axes on the country/cluster specific time-series plots to match the time series from Figure 2. I don’t feel strongly about this, but for the sake of historical context, dates may be more useful than indexed weeks.

- Line 498: missing punctuation after “synchrony”

- Lines 549-551: awkward sentence structure should be reworked (maybe one sentence got broken into two?)

Discussion

- Line 620: typo “dcay”

- Line 661: missing word “allows us to understand”

Reviewer #2: In this work, the authors use Poisson and Hawkes processes to model the temporal dynamics of pandemic-related disorder events in India, Israel, and Mexico, and use k-means clustering to assess spatial clustering. The manuscript is well-written and appears scientifically sound. However, the manuscript would benefit from more thorough citation, more condensed figures, and better consistency across figures. I recommend that the manuscript be accepted contingent on addressing these issues.

Major comments

1. The introduction would benefit from more citations for various claims about the global response to COVID-19 and the role of media, if such references are available. I don’t doubt that these claims are accurate, but think the introduction would be strengthened by the addition of more references.

2. Have similar methods been used to study other types of disorder, violence, or other human behaviors? Please add a brief statement with references in the introduction on the applicability of these methods to this type of problem. It looks like examples are given in Section 2, but it is not clear whether the methodology is similar.

3. Are there references for the estimates of what proportion of the population had internet access in 2020?

4. Section 2.1 would also benefit from additional references for the specific examples that are given, such as: protests in response to the CAA; traffic light monitoring plans; clashes with police; increase in murders and femicides.

5. Please include a citation for the COVID-19 Disorder Tracker dataset and ACLED in Section 3 (line 270-271).

6. Figure 2 seems to be missing.

7. Line 417: “Cluster 2 accounts for the majority of disorder events” – this is not quite accurate, and the term “plurality” should be used instead.

8. Tables 2, 3, 4: Are there tests for statistical significance that can be used to compare these values between clusters?

9. Line 463: by “more damped” feedback, do you mean shorter? This is not totally clear; perhaps the wording should be “more quickly damped”.

10. Can differences in the number of events for each cluster be explained primarily by population size? It seems natural that more events would occur in larger populations.

11. In Figure 10, it is misleading to use the scale of 0.95-1 since there is really very little difference in values, yet visually it appears there are large differences. This figure could likely even be omitted, since there is no meaningful difference between the correlations. Similarly, in Figure 6 it is misleading to use a two-color scale when all values are positive. In general, for the pairwise correlation plots, there should be a consistent color map from -1 to 1, with one color for negative values and another color for positive values, so that the color scale is the same across all plots. This would allow for comparison between plots and better interpretation.

12. There are quite a lot of figures. It may be beneficial to condense some figures into panel figures; for example, to combine all figures for one country into a single panel figure.

Minor comments

13. Line 58: “Disorded” -> “Disorder”

14. Line 164: “22th” -> “22nd”

15. Line 244: “began reported” -> “began reporting”

16. The figure files don’t seem to be presented in order, and there are multiple files for each figure. Please ensure figures are correctly inserted for publication.

6. PLOS authors have the option to publish the peer review history of their article (what does this mean?). If published, this will include your full peer review and any attached files.

Reviewer #1: No

Reviewer #2: No

---

## [Author Response · Author response to Decision Letter 0]

12 Mar 2021

Dear Editor,

Dear Reviewers,

We are grateful for the time spent on our manuscript and the insightful comments provided. We have addressed them one by one, and we feel that, after having taken care of your suggestions, the manuscript has been significantly improved in both its style and contents.

As anticipated, please find below our point-by-point response.

We modified file naming and other stylistic choices to match the requirements listed in the PLOS ONE style templates. While we believe all requirements are now met, we kindly ask the Editor to let us know whether further attention needs to be devoted to formatting issues.

2. We note that Figures 3, 7 and 12 in your submission contain map/satellite images which may be copyrighted.

a. You may seek permission from the original copyright holder of Figures 3, 7 and 12 to publish the content specifically under the CC BY 4.0 license.

We appreciate the attention to proper licensing of figures 4, 7, and 10. Google Maps explicitly allows use of their materials for research purposes, provided it is fair use and there is proper attribution. Please see https://about.google/brand-resource-center/products-and-services/geo-guidelines/. Ours is a scientific publication, justifying fair use. In the revised version of the manuscript, and as required by Google Maps, we have provided proper attribution in each figure caption and in the Acknowledgments.

We have now added a caption for the Supporting Information file at the end of the manuscript and checked in-text citations.

Reviewers' comments:

Reviewer's Responses to Questions

Comments to the Author

1. Is the manuscript technically sound, and do the data support the conclusions?

Reviewer #1: Yes

Reviewer #2: Yes

2. Has the statistical analysis been performed appropriately and rigorously?

Reviewer #1: Yes

Reviewer #2: Yes

3. Have the authors made all data underlying the findings in their manuscript fully available?

Reviewer #1: Yes

Reviewer #2: Yes

4. Is the manuscript presented in an intelligible fashion and written in standard English?

Reviewer #1: Yes

Reviewer #2: Yes

5. Review Comments to the Author

Reviewer #1: This manuscript presents an interesting application of Hawkes processes and k-means clustering to the spatial and temporal relationships between civil disorder events. In particular, the paper provides a quantitative evaluation of the suspected interplay between a government’s efforts to control the COVID-19 pandemic and resultant civil unrest. I do not have any major comments on the use of Hawkes processes for this data. I found the description of the method and communication of results very informative.

Major Comments:

Background

- Line 189+: When the BLM protests are discussed earlier in the paper, it’s stated that disorder such as this would not be included in the work since they “arise primarily in response to racial injustice, although racial disparities in COVID-19 cases may have contributed to the unrest.” This seems like a sensible exclusion, and I wonder if this exclusion criteria should also apply to the June CAA demonstrations. The authors seem to make the case that these are not primarily a result of the pandemic. In fact, they seem to be quelled rather than incited following lockdown orders.

We thank the reviewer for giving us the opportunity to clarify this point. We confirm that social unrest events concurrent with the pandemic are not tallied by the ACLED’s COVID Disorder Tracker initiative, and therefore are not included in our analyses. What is reported in the Background section (particularly in reference to India), is simply a description of the timeline of the most relevant events that took place during 2020, at the same time of the pandemic. This does not mean that CAA disorders are part of the ACLED/CDT dataset. If they were, we would have excluded them from our analyses, given that – as for the BLM protests – their motive is not intrinsically related to the pandemic, as the reviewer rightly points out.

ACLED’s methodology in filtering out non-pandemic related protests is very clearly detailed in the brief published (and cited in our manuscript) at https://acleddata.com/acleddatanew/wp-content/uploads/dlm_uploads/2020/04/ACLED_Direct-COVID19-Disorder_Methodology-Brief_4.2020.pdfcontent/uploads/dlm_uploads/2020/04/ACLED_Direct-COVID19-Disorder_Methodology-Brief_4.2020.pdf

where they state:

For users interested in data capturing disorder linked to the coronavirus, ACLED offers a curated data file containing all events that are directly related to the pandemic. By ‘directly related’, we mean all events for which the word ‘coronavirus’ is included in the ‘Notes’ column of the ACLED dataset to specifically indicate that the pandemic was an explicit factor. ACLED Researchers include this tag only when an incident report makes obvious that the coronavirus motivated the event (e.g. “A protest occurred in opposition to movement restrictions imposed as a result of the coronavirus”). Events that are directly related to the pandemic include things such as:

● The targeting of healthcare workers responding to the coronavirus

● Violent mobs attacking individuals due to fears of their alleged links to the coronavirus (e.g. Muslims in India; foreigners in Africa; etc.)

● Demonstrations against governance decisions made in response to the coronavirus

No reference to the coronavirus will be made for events that are not directly related to the pandemic (e.g. “A battle occurred and was not connected to the coronavirus”). In this way, ACLED can ensure that the ‘coronavirus’ tag does not trigger such an event’s inclusion in the curated data file as a false positive

Results

- I wonder if the box plots contribute enough to the overall objective to remain included as figures or if they could be removed in order to avoid detracting from the more informative results. There is very little main text dedicated to interpreting them in the Results section and none in the Discussion. Given the expected (and demonstrated) skew of the data, it seems like a less than ideal visualization technique. If the average count of weekly events is worth reporting explicitly, then that can be accomplished in the text and without a corresponding figure. Further, the distribution of weekly counts and its variability can be more clearly observed in the time series plots, which also allows the reader to see the temporal trends that lead up to "outlier" weeks.

We agree with the reviewers’ comment regarding the marginal usefulness of the box plot, especially compared to the time-series plots. In light of this, we have now moved the box plots to the Supporting Information.

Minor Comments:

Background

- Generally, these paragraphs are very dense with important and overlapping dates for each country. Perhaps a timeline figure would complement these descriptions and make this information easier to parse.

We have now provided a figure at the end of the Background section illustrating a timeline with the most relevant events in the COVID-19 pandemic for each country.

- Line 159-160: awkward sentence structure ?

Fixed.

- Line 165: “22nd”

Fixed.

- Line 168-169: awkward sentence structure

Corrected.

- Line 254: “including those who work in the informal economy”

Fixed.

Materials and Methods

- Fig. 1 caption:

o PLOS One guidelines require that all non-standard abbreviations be defined. Would recommend for “Violence a.c.”

We have now added a definition in the caption.

o There’s a spelling issue in the database name (should be ACLED not ALCED)

Fixed. Thanks for notifying us.

o Consider including the date range for the data displayed to make the figure and caption stand alone.

Included.

Results

- Consider changing the x-axes on the country/cluster specific time-series plots to match the time series from Figure 2. I don’t feel strongly about this, but for the sake of historical context, dates may be more useful than indexed weeks.

We have carefully considered this option, but have concluded that for clarity it is best to keep the axes as they are. In Figure 2 we focus on time-series at the daily level and plot the data jointly for all three countries of interest. In this case, using actual dates facilitates the processing and interpretation of the graphic. However, in the country/cluster specific time-series we display weekly-level time series for each country without juxtaposing data for all three so it makes more sense to keep indexed weeks which are easier to display and which reduce the risk of confusion. Note that we do provide information on start and end dates, to allow for the extrapolation of actual dates from the indexed ones. Furthermore, when discussing the findings from these plots in the main text, we included the corresponding actual dates. This should help the reader better place events within the timeline of the pandemic.

- Line 498: missing punctuation after “synchrony”

Fixed.

- Lines 549-551: awkward sentence structure should be reworked (maybe one sentence got broken into two?)

Fixed. Thank you again for pointing this out.

Discussion

- Line 620: typo “decay”

- Line 661: missing word “allows us to understand”

Both fixed.

Reviewer #2: In this work, the authors use Poisson and Hawkes processes to model the temporal dynamics of pandemic-related disorder events in India, Israel, and Mexico, and use k-means clustering to assess spatial clustering. The manuscript is well-written and appears scientifically sound. However, the manuscript would benefit from more thorough citation, more condensed figures, and better consistency across figures. I recommend that the manuscript be accepted contingent on addressing these issues.

Major comments

1. The introduction would benefit from more citations for various claims about the global response to COVID-19 and the role of media, if such references are available. I don’t doubt that these claims are accurate, but think the introduction would be strengthened by the addition of more references.

We have now provided citations in the Introduction addressing the role of the media, as well as other ones we believe deserved better referencing. The references added in the Introduction are:

1. Al-Dmour, H., Masa’deh, R., Salman, A., Abuhashesh, M., & Al-Dmour, R. (2020). Influence of Social Media Platforms on Public Health Protection Against the COVID-19 Pandemic via the Mediating Effects of Public Health Awareness and Behavioral Changes: Integrated Model. Journal of Medical Internet Research, 22(8), e19996. https://doi.org/10.2196/19996

2. Alfano, V., & Ercolano, S. (2020). The Efficacy of Lockdown Against COVID-19: A Cross-Country Panel Analysis. Applied Health Economics and Health Policy, 18(4), 509–517. https://doi.org/10.1007/s40258-020-00596-3

3. Brady, R. R., Insler, M., & Rothert, J. (2020). The Fragmented United States of America: The Impact of Scattered Lock-Down Policies on Country-Wide Infections (SSRN Scholarly Paper ID 3681486). Social Science Research Network. https://doi.org/10.2139/ssrn.3681486

4. Chen, Q., Min, C., Zhang, W., Wang, G., Ma, X., & Evans, R. (2020). Unpacking the black box: How to promote citizen engagement through government social media during the COVID-19 crisis. Computers in Human Behavior, 110, 106380. https://doi.org/10.1016/j.chb.2020.106380

5. Delen, D., Eryarsoy, E., & Davazdahemami, B. (2020). No Place Like Home: Cross-National Data Analysis of the Efficacy of Social Distancing During the COVID-19 Pandemic. JMIR Public Health and Surveillance, 6(2), e19862. https://doi.org/10.2196/19862

6. Kouzy, R., Abi Jaoude, J., Kraitem, A., El Alam, M. B., Karam, B., Adib, E., Zarka, J., Traboulsi, C., Akl, E. W., & Baddour, K. (2020). Coronavirus Goes Viral: Quantifying the COVID-19 Misinformation Epidemic on Twitter. Cureus, 12(3). https://doi.org/10.7759/cureus.7255

7. Mills, M. C., & Salisbury, D. (2021). The challenges of distributing COVID-19 vaccinations. EClinicalMedicine, 31. https://doi.org/10.1016/j.eclinm.2020.100674

8. Muriel, J.-J., & Bauchner, H. (2021). Vaccine Distribution—Equity Left Behind? JAMA. https://doi.org/10.1001/jama.2021.1205

9. Pennycook, G., McPhetres, J., Zhang, Y., Lu, J. G., & Rand, D. G. (2020). Fighting COVID-19 Misinformation on Social Media: Experimental Evidence for a Scalable Accuracy-Nudge Intervention. Psychological Science, 31(7), 770–780. https://doi.org/10.1177/0956797620939054

10. Tasnim, S., Hossain, M. M., & Mazumder, H. (2020). Impact of Rumors and Misinformation on COVID-19 in Social Media. Journal of Preventive Medicine and Public Health, 53(3), 171–174. https://doi.org/10.3961/jpmph.20.094

11. Wellenius, G. A., Vispute, S., Espinosa, V., Fabrikant, A., Tsai, T. C., Hennessy, J., Dai, A., Williams, B., Gadepalli, K., Boulanger, A., Pearce, A., Kamath, C., Schlosberg, A., Bendebury, C., Mandayam, C., Stanton, C., Bavadekar, S., Pluntke, C., Desfontaines, D., … Gabrilovich, E. (2020). Impacts of US State-Level Social Distancing Policies on Population Mobility and COVID-19 Case Growth During the First Wave of the Pandemic. ArXiv:2004.10172 [q-Bio]. http://arxiv.org/abs/2004.10172

2. Have similar methods been used to study other types of disorder, violence, or other human behaviors? Please add a brief statement with references in the introduction on the applicability of these methods to this type of problem. It looks like examples are given in Section 2, but it is not clear whether the methodology is similar.

In the revised manuscript we have included a discussion and multiple references on the use of Hawkes processes in understanding patterns of violent human phenomena. These are:

1. Brantingham, P. J., Yuan, B., & Herz, D. (2020). Is Gang Violent Crime More Contagious than Non-Gang Violent Crime? Journal of Quantitative Criminology. https://doi.org/10.1007/s10940-020-09479-1

2. Loeffler, C., & Flaxman, S. (2018). Is Gun Violence Contagious? A Spatiotemporal Test. Journal of Quantitative Criminology, 34(4), 999–1017. https://doi.org/10.1007/s10940-017-9363-8

3. Tench, S., Fry, H., & Gill, P. (2016). Spatio-temporal patterns of IED usage by the Provisional Irish Republican Army. European Journal of Applied Mathematics, 27(3), 377–402.

3. Are there references for the estimates of what proportion of the population had internet access in 2020?

We have now added the requested references for each country.

4. Section 2.1 would also benefit from additional references for the specific examples that are given, such as: protests in response to the CAA; traffic light monitoring plans; clashes with police; increase in murders and femicides.

We now provide references related to the abovementioned events as well as to other important pandemic-related facts in the three countries of interest. These references are:

1. Jaffe-Hoffman, M. (2020). Red light, green light, go! Gamzu’s traffic light plan passes. The Jerusalem Post. https://www.jpost.com/health-science/coronavirus-cabinet-to-convene-sneak-peek-at-gamzus-traffic-light-plan-640464

2. Murray, O. L., Christine. (2020, April 27). Murders of women in Mexico rise amid fears of lockdown violence. Reuters. https://www.reuters.com/article/us-mexico-women-violence-trfn-idUSKCN22930V

3. Pokharel, K., & Purnell, N. (2019). Protests Over India’s New Citizenship Law Widen. The Wall Street Journal. https://www.wsj.com/articles/protests-over-indias-new-citizenship-law-widen-11576501527

4. Sanchez, D. J., Uriel. (2020, June 18). As Mexico focuses on coronavirus, drug gang violence rises. Reuters. https://www.reuters.com/article/uk-health-coronavirus-mexico-cartels-idUKKBN23P1T5

5. Shpigel, N., Peleg, B., Hasson, N., Breiner, J., & Shezaf, H. (2020). Clashes, arrests as hundreds of anti-Netanyahu protests held across Israel under lockdown. Haaretz. https://www.haaretz.com/israel-news/.premium-despite-restrictions-anti-netanyahu-protests-continue-at-thousands-of-locations-1.9205689

5. Please include a citation for the COVID-19 Disorder Tracker dataset and ACLED in Section 3 (line 270-271).

We have now added the requested citation.

6. Figure 2 seems to be missing.

We have now included the figure.

7. Line 417: “Cluster 2 accounts for the majority of disorder events” – this is not quite accurate, and the term “plurality” should be used instead.

We have corrected as suggested.

8. Tables 2, 3, 4: Are there tests for statistical significance that can be used to compare these values between clusters?

To the best of our knowledge, not quite. The Kolmogorov-Smirnov (KS) test is typically used as a goodness of fit test for Hawkes processes, and implies that the set of compensators calculated at each event time should define a Poisson processes. It may be that two clusters differ in the outcomes of the KS test (e.g., one of them passes the KS test the other does not), and therefore we can “compare” the clusters claiming that in one case there is support for the use of a Hawkes process with an exponential decay function, while in the other the modeling choice does not fit well. However, it would make little sense to compare the magnitude and significance of the tests between clusters, especially given that the sampled residuals vary in numbers across clusters.

9. Line 463: by “more damped” feedback, do you mean shorter? This is not totally clear; perhaps the wording should be “more quickly damped”

Thanks for pointing this out. We have clarified by using “more quickly damped”, as suggested.

10. Can differences in the number of events for each cluster be explained primarily by population size? It seems natural that more events would occur in larger populations.

We appreciate very much the reviewer's suggestion to correlate the number of protes events in each cluster with its population. In the paper, we identified the clusters with the largest number of events, however, due to the many possible variables at play, thoroughly correlating the number of events with cluster-specific characteristics (not just density, but also overall population, percent of rural population, percent of young persons, income, climate, political leanings, government responsiveness) would require an analysis of its own, and is outside the scope of this paper.

Nonetheless, during this review process, we did take a closer look of the data, and noted that in India the greatest number of events does often occur in clusters that contain populous or densely populated states or Union Territories (e.g., Punjab and Assam in Clusters 1 and 2, respectively). However, the Union Territory with the highest number of protest events - Jammu and Kashmir - is neither among the most populous or the densest, implying that other socio-political or socio-economic factors might be at play in igniting the covid-related protests.

The data from Mexico shows that the greatest number of events occurs in Cluster C4 that contains the most populous, dense states of the country including the capital, Mexico City (population 9 million, density 6,180 persons/km2), and the state of Mexico proper.

Finally, in Israel, the greatest number of events occurs in Cluster C4 that contains the city of Haifa, among the most densely populated in Israel. Cluster 3, which is the second highest in terms of recorded events, contains other populated territories of the country, including Tel Aviv and the Sharon plain. Interestingly, Jerusalem, also one of the highest density areas in the country, is not characterized by a significantly high number of disorder events.

As mentioned above, these not-so-clear cut observations show that correlating the number of events with specific cluster demographic or geographical characteristics may be too simplistic. For completeness, we added a paragraph in the conclusion with a brief discussion, underlying the need to consider the intersectionality of socioeconomic, political and/or religious factors when trying to understand why some clusters display more events than others.

11. In Figure 10, it is misleading to use the scale of 0.95-1 since there is really very little difference in values, yet visually it appears there are large differences. This figure could likely even be omitted, since there is no meaningful difference between the correlations. Similarly, in Figure 6 it is misleading to use a two-color scale when all values are positive. In general, for the pairwise correlation plots, there should be a consistent color map from -1 to 1, with one color for negative values and another color for positive values, so that the color scale is the same across all plots. This would allow for comparison between plots and better interpretation.

We thank the reviewer for this comment, which allowed us to rethink the way we present our correlation plots in the manuscript at the subnational scale and to find ways to better display our findings. We believe that the correlation plots are important as they convey relevant information regarding the event distributions within each cluster. Hence, we think it is more informative to keep them in the main body of the manuscript.

To strike a balance between plot comparability (i.e., one should be able to compare each plot in each country with the corresponding plot in another country) and consistency (i.e. the same number and types of figures should be displayed for each country) we partially followed the reviewer’s advice. Specifically, in the revised version of the manuscript, the correlation plots for all three countries are produced using the same color map, aka the Spectral palette in ggplot. Furthermore, the plots for India and Mexico are drawn using the entire (-1;+1) range, as typical of Pearson’s correlation plots. These two choices facilitate comparability across countries. However, we maintained a restricted scale (+0.95; +1) for Israel to allow for more insight, given the high values of the correlation coefficient nationwide.

The rationale behind this choice is better illustrated in the caption of Figure 9 in the revised manuscript where the differing range for Israel compared to India and Mexico is specifically pointed to the reader’s attention. We preferred this choice compared to the other two alternatives, (a) deleting the figure for Israel, reducing consistency across countries and (b) maintaining the original scale (-1,+1), which would have led to a uniformly colored, and not informative, plot.

12. There are quite a lot of figures. It may be beneficial to condense some figures into panel figures; for example, to combine all figures for one country into a single panel figure.

To address both this comment and another comment from the other reviewer, we have now modified the figures removing the box plots for each country, thus reducing the number of plots overall.

Minor comments

13. Line 58: “Disorded” -> “Disorder”

Fixed.

14. Line 164: “22th” -> “22nd”

Fixed.

15. Line 244: “began reported” -> “began reporting”

Fixed. Thank you for spotting these typos.

16. The figure files don’t seem to be presented in order, and there are multiple files for each figure. Please ensure figures are correctly inserted for publication.

We have verified that the figures are presented in the correct order in the revised manuscript.

---

## [Decision Letter · Decision Letter 1]

7 Apr 2021

Temporal Clustering of Disorder Events During the COVID-19 Pandemic

PONE-D-21-01644R1

Dear Dr. Campedelli,

We’re pleased to inform you that your manuscript has been judged scientifically suitable for publication and will be formally accepted for publication once it meets all outstanding technical requirements.

Kind regards,

Chad M. Topaz

Academic Editor

PLOS ONE

Additional Editor Comments (optional):

Reviewers' comments:

Reviewer's Responses to Questions

**Comments to the Author**

1. If the authors have adequately addressed your comments raised in a previous round of review and you feel that this manuscript is now acceptable for publication, you may indicate that here to bypass the “Comments to the Author” section, enter your conflict of interest statement in the “Confidential to Editor” section, and submit your "Accept" recommendation.

Reviewer #1: All comments have been addressed

Reviewer #2: All comments have been addressed

2. Is the manuscript technically sound, and do the data support the conclusions?

Reviewer #1: Yes

Reviewer #2: Yes

3. Has the statistical analysis been performed appropriately and rigorously? 

Reviewer #1: Yes

Reviewer #2: Yes

4. Have the authors made all data underlying the findings in their manuscript fully available?

Reviewer #1: Yes

Reviewer #2: Yes

5. Is the manuscript presented in an intelligible fashion and written in standard English?

Reviewer #1: Yes

Reviewer #2: Yes

6. Review Comments to the Author

Reviewer #1: (No Response)

Reviewer #2: (No Response)

7. PLOS authors have the option to publish the peer review history of their article (what does this mean?). If published, this will include your full peer review and any attached files.

Reviewer #1: No

Reviewer #2: No

---

## [Editor Report · Acceptance letter]

13 Apr 2021

PONE-D-21-01644R1

Temporal Clustering of Disorder Events During the COVID-19 Pandemic

Dear Dr. Campedelli:

I'm pleased to inform you that your manuscript has been deemed suitable for publication in PLOS ONE. Congratulations! Your manuscript is now with our production department.

Kind regards,

on behalf of

Dr. Chad M. Topaz

Academic Editor

PLOS ONE